# Evidence accumulation relates to perceptual consciousness and monitoring

Michael Pereira [1,2,3], Pierre Megevand[4,5,6], Mi Xue Tan [1], Wenwen Chang[1], Shuo Wang [3,7], Ali Rezai[3], Margitta Seeck[4], Marco Corniola[8,9], Shahan Momjian[8,9], Fosco Bernasconi[1], Olaf Blanke[1,9,10] & Nathan Faivre [1,2,10 ✉]

A fundamental scientific question concerns the neural basis of perceptual consciousness and perceptual monitoring resulting from the processing of sensory events. Although recent studies identified neurons reflecting stimulus visibility, their functional role remains unknown. Here, we show that perceptual consciousness and monitoring involve evidence accumulation. We recorded single-neuron activity in a participant with a microelectrode in the posterior parietal cortex, while they detected vibrotactile stimuli around detection threshold and provided confidence estimates. We find that detected stimuli elicited neuronal responses resembling evidence accumulation during decision-making, irrespective of motor confounds or task demands. We generalize these findings in healthy volunteers using electroencephalography. Behavioral and neural responses are reproduced with a computational model considering a stimulus as detected if accumulated evidence reaches a bound, and confidence as the distance between maximal evidence and that bound. We conclude that gradual changes in neuronal dynamics during evidence accumulation relates to perceptual consciousness and perceptual monitoring in humans.

[1] Laboratory of Cognitive Neuroscience, Center for Neuroprosthetics and Brain Mind Institute, Faculty of Life Sciences, Swiss Federal Institute of Technology (EPFL), Geneva, Switzerland. [2] University Grenoble Alpes, University Savoie Mont Blanc, CNRS, LPNC, Grenoble, France. [3] Rockefeller Neuroscience Institute (RNI), West Virginia University, Morgantown, USA. [4] Neurology Division, Department of Clinical Neuroscience, Geneva University Hospitals, Geneva, Switzerland. [5] Department of Fundamental Neuroscience, University of Geneva, Geneva, Switzerland. [6] Wyss Center for Bio and Neuroengineering, Geneva, Switzerland. [7] Department of Chemical and Biomedical Engineering, West Virginia University, Morgantown, USA. [8] Neurosurgery Division, Department of Clinical Neuroscience, University of Geneva University Hospitals, Geneva, Switzerland. [9] Faculty of Medicine, University Hospital Geneva, Geneva, Switzerland. [10] These authors contributed equally: Olaf Blanke, Nathan Faivre ✉email: nathanfaivre@gmail.com

The processing of sensory signals by the human brain gives rise to two interrelated phenomena: perceptual consciousness, defined as the subjective experience associated with a sensory event[1,2], and perceptual monitoring, defined as the capacity to introspect and reflect upon the subjective experience associated with a sensory event[3,4]. The main strategy employed to study perceptual consciousness consists in relating first-order subjective reports to neural activity to identify the minimal set of neuronal events and mechanisms sufficient for a specific conscious percept (i.e., neural correlates of consciousness or NCCs[5]). To identify NCCs, most experimental paradigms have adopted a contrastive approach, whereby distinct phenomenal experiences induced by constant sensory stimulation are compared[6]. One of the simplest contrasts is obtained when stimuli are presented at low intensity or embedded in noise so that only a certain proportion of them is detected[7]. A comparison of neural activity elicited by detected and missed stimuli allows distinguishing the neural correlates of conscious vs. unconscious sensory processing, and therefore identifying NCCs given that specific confounds are ruled out[8]. However, although rare investigations in humans have described single neurons in the temporal lobe encoding stimulus detection[9,10], the mechanistic role of neuronal activity for perceptual consciousness remains unknown. One prominent theory of consciousness, the global neuronal workspace, proposes that a stimulus is consciously perceived when its corresponding neural activity is globally broadcasted across the cortex[11]. This theory assumes that this global broadcast is triggered when an (unconscious) evidence accumulation process reaches a threshold[12] similar to the physiological processes underlying decision-making[13–15]. A behavioral modeling study proposed a similar mechanism for the subjective experience of reaching a decision[16]. Various neuroimaging studies have interpreted increases in neural activity elicited by detected stimuli[17] (versus missed stimuli) as evidence accumulation[18–20] and several animal studies have reported single neuron correlates in detection tasks[15,21,22]. However, little empirical evidence supports an evidence accumulation account of perceptual consciousness, especially at the single neuron level.

Besides perceptual consciousness, the main strategy to study perceptual monitoring consists in assessing how second-order reports like confidence judgments co-vary with the accuracy of a given perceptual task[23]. As most studies investigating perceptual monitoring rely on first-order discrimination tasks with stimuli that are always detected, less is known regarding how the brain monitors the presence or absence of subjective experience[24,25]. Moreover, the interdependencies between perceptual consciousness and monitoring remain to be described empirically: while some theories argue that consciousness requires a higher order representation of a stimulus[26,27], or necessarily comes with a sense of confidence[28], other theories argue that first-order representations may be sufficient for a conscious percept to occur[29,30]. Like for perceptual consciousness, several models propose that evidence accumulation plays an important role for the formation of perceptual confidence[31–34]. Yet, to our knowledge the underlying neural mechanisms of confidence in detection tasks remain to be described.

Here, we sought to investigate the role of evidence accumulation in perceptual consciousness and perceptual monitoring by asking participants to detect weak vibrotactile stimuli and rate their confidence in having detected them. We reasoned that both detection and confidence underlie decision-making processes whereby participants accumulate perceptual evidence over time and gauge its level relative to decision criteria. We examined this possibility in a patient implanted with a micro-electrode array in the posterior parietal cortex (PPC, Fig. 1a), considered as one of the functional hotspots of evidence accumulation in the non-human primate brain[13,14]. We isolated 368 putative single neurons (Supplementary Fig. 1) in three different experiments with immediate responses (Experiment 1), delayed responses (Experiment 2, 4), and no-report (Experiment 3) in order to characterize the neural correlates of detection and confidence at the single-neuron and population levels and link evidence accumulation in the PPC to perceptual consciousness and monitoring. Namely, the comparison of neural signals between paradigms involving immediate and delayed responses ensured that the NCCs we isolated were not contaminated by motor actions. Moreover, the use of a no-report paradigm ensured that these NCCs reflected perceptual consciousness per se, irrespective of task demand[35]. These results were generalized in a fourth experiment involving a group of healthy volunteers in whom we recorded scalp electroencephalography, perceptual consciousness and monitoring responses while they detected the same vibrotactile stimuli. In a final step, we test and propose an evidence accumulation computational model that reproduced the behavioral and neural markers of both detection and confidence. Together, these results indicate that subjective reports of perceptual consciousness and monitoring involve a common mechanism of evidence accumulation orchestrated by the PPC.

## Results

**Experiment 1: immediate-response task.** In Experiment 1, the participant was asked to detect vibrotactile stimuli applied to the right wrist (contralateral to the PPC implant) with an intensity around detection threshold. Responses were provided by a keypress with the left hand, immediately after perceiving a stimulus. A trial was considered a hit when the participant responded within 2 s following stimulus onset (41.20% of trials; mean response time (RT) and 95% confidence interval: $0.71 \pm 0.02$s; Fig. 1c), otherwise, it was considered a miss (58.80% of trials Fig. 1b). The participant rarely responded outside this 2 s window (0.36%; false-alarms), indicative of conservative behavior. We found 81/186 detection-selective neurons (43.55%; Supplementary Fig. 2a; $p = 0.001$, Wilcoxon rank test with permutation test across neurons) with spike counts explained by detection (yes/no responses) between 0.5 and 1.5 s after the stimulus onset. 44/81 of these detection-selective neurons significantly increased their firing rate compared to a 300 ms baseline prior to stimulus onset (Supplementary Fig. 2b; $p = 0.001$) while only one decreased its firing rate ($p = 0.78$). Crucially, some neurons were characterized by a hallmark of evidence accumulation with gradually increasing firing rates prior to the keypress; to find such neurons, we tested for correlations between RT and the slope of the firing rates between 300 ms and the response onset. We found 14/81 such RT-selective neurons (17.28%; $p = 0.001$; Fig. 1d–f; see Supplementary Fig. 3 for a confirmatory analysis). Consistent with evidence accumulation, highest correlations were negative and related to positive choice probabilities (Fig. 1g).

Regarding misses, 16/81 neurons significantly increased their firing rate compared to baseline (Supplementary Fig. 2c; $p = 0.001$). Average firing rates for these 16 neurons suggested that this increase from baseline, was lower than the one we observed for hits. We therefore sought for neurons that increased their firing rates for misses and had higher spike counts for misses than for hits. We found six such miss-coding neurons (Fig. 1g, Supplementary Fig. 2d; $p = 0.001$), suggesting that some neurons in the PPC actively code missed stimuli[22]. Finally, we found qualitatively similar increases in firing rate with electrocorticography (ECoG) with an effect of detection that was strongest in the PPC and pre-central gyrus (Fig. 1h–i). To summarize, we uncovered individual neurons in the human PPC with firing rates

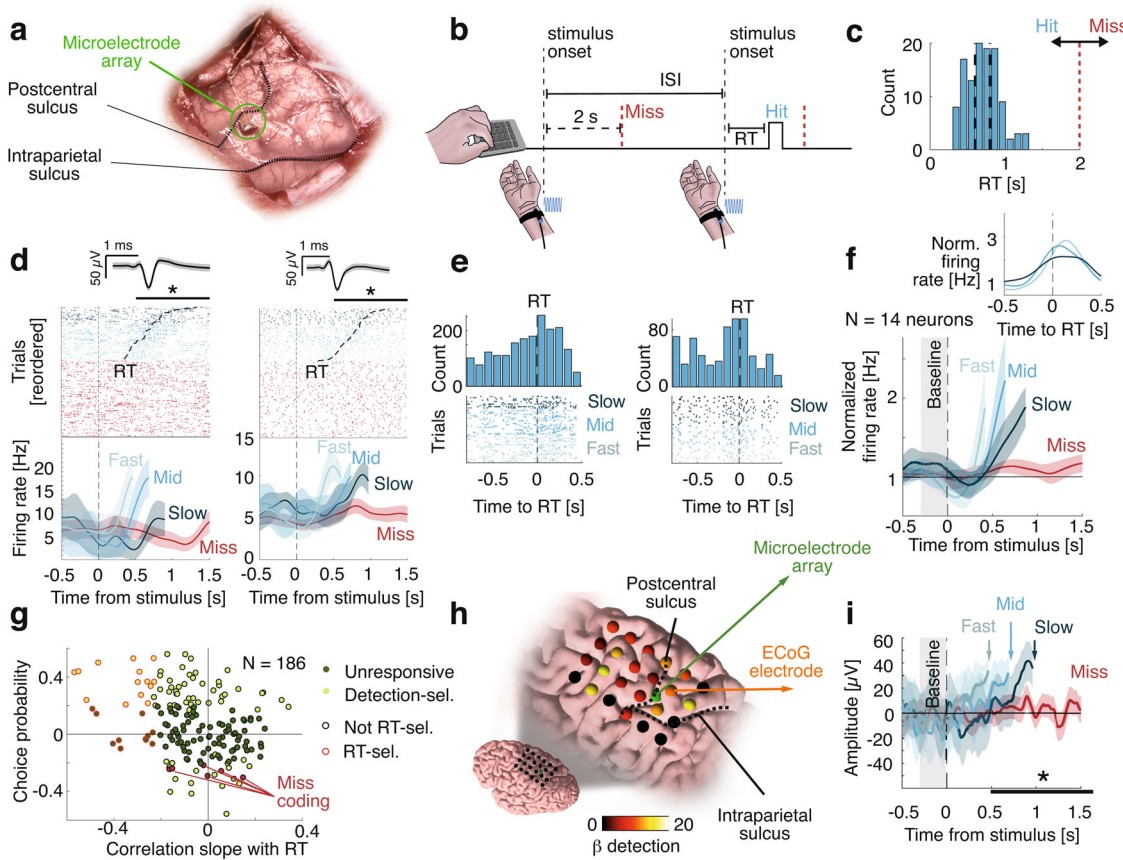

**Fig. 1 Neuronal correlates of detection in an immediate-response task (Experiment 1). a** Intraoperative photo of the microelectrode array posterior to the postcentral sulcus and dorsal to the intraparietal sulcus. **b** The participant pressed a key as soon as they felt a stimulus. In this example, the first stimulus is a miss (i.e., no key press within 2 s following stimulus onset) and the second stimulus is a hit (i.e., key press within 2 s following stimulus onset). ISI inter-stimulus interval. **c** Distribution of response times (RT) and response deadline for a trial to be considered as a hit (red vertical dashed line). The y axis represents the number of trials. **d** Example detection- and RT-selective neurons. Top: raster plot time-locked to stimulus onset with spike waveform and shaded standard deviation above. Hits were reordered according to RT (black dashed trace). Bottom: average firing rate for three terciles of RT for hits and for misses. Statistics were performed on continuous RT data. Asterisks indicate statistical significance; left: detection-selective: $p = 0.013$ (Wilcoxon ranksum test), RT-selective: $p < 0.001$ (Spearman correlation); Right: detection-selective: $p = 0.046$, RT-selective: $p = 0.0056$ **e** Top: RT-aligned spike count histograms for neurons in **d** (50 ms bins). The y axis represents the number of trials. Bottom: corresponding raster plots. **f** Average firing rates for detection- and RT- responsive neurons, for three bins of RT for hits and for misses. Firing rates were normalized to a 0.3 s pre-stimulus baseline and time-locked to the stimulus onset. Inset: Corresponding average firing rates time-locked to the response. **g** Choice probability as a function of the correlation between the slope of the firing rate and the RT for all neurons with detection-selective neurons in light green, miss coding in red and other neuron in dark green. RT-selective neurons are circled with red. **h** ECoG grid with beta (β) coefficients for detection. Non-significant electrodes are in black. **i** Average ECoG amplitude, aligned to stimulus onset from one electrode posterior to the microelectrode array for three terciles of RT for hits and for misses. Asterisk indicates statistical significance; $p < 0.001$ (linear model). In all panels, misses are depicted in red and hits in blue, shaded areas represent 95%-CI, black horizontal bars represent the analysis window for statistics.

ramping up prior to detection reports, consistent with evidence accumulation.

**Experiment 2: delayed-response task.** Because the participant hit a key as soon as they detected a stimulus in Experiment 1, the comparison between hits and misses mentioned above may be contaminated by motor confounds. Thus, we tested whether neuronal responses relate to conscious perception irrespective of motor actions by imposing a delay between stimulus onset and reports. In Experiment 2, we asked the participant to report vocally the detection of the stimuli with a minimal delay of 1 s after stimulus onset (Fig. 2a, upper panel). To assess the role of evidence accumulation for perceptual monitoring, we also asked the participant to vocally report their confidence (high, medium, low) in their response[36,37]. Similar to Experiment 1, 50.5% of stimuli were detected (hits) (20% trials had no stimuli (catch

trials), of which 5% were false-alarms, confirming their conservative strategy). When a stimulus was presented, confidence was higher following hits ($2.46 \pm 0.10$) than misses ($2.00 \pm 0.08$; $X^2 = 20.09$, $p = 4.3*10-5$; Fig. 2a, lower panel).

We ran a factorial analysis to identify neurons encoding detection and/or confidence. Only one neuron showed only a main effect of detection (1.16%, $p = 0.86$) and two a main effect of confidence (2.33%, $p = 0.97$). However, 17/86 neurons showed an interaction between detection and confidence (19.77%, $p = 0.02$, permutation test) driven by firing rates for hits with high confidence (Fig. 2b). Among these 17 neurons, we found nine neurons showing a significant correlation between spike counts and confidence in hits ($p = 0.001$). The sign of this correlation was consistent with the choice probability in eight of these nine neurons, suggesting that evidence for detection increases confidence in hits. We found no miss-coding neuron, and no

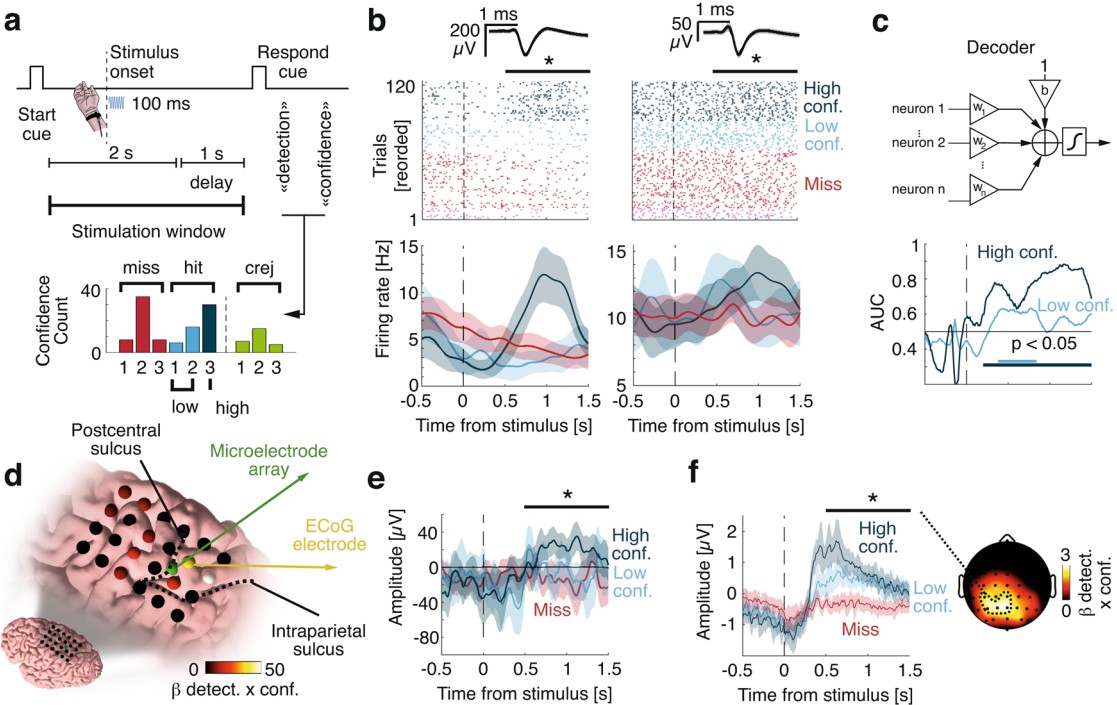

**Fig. 2 Neuronal correlates of detection and confidence in a delayed-response task (Experiment 2). a** Top: Vibrotactile stimuli were applied during a 2 s window following an auditory cue. After a 1 s delay, the participant was prompted to give detection and confidence reports. Bottom: Distribution of confidence. For display purposes hereafter, signals corresponding to confidence values of 1 and 2 were merged into low confidence, while confidence values of 3 were considered as high-confidence. Statistics were done on the three levels. **b** Example selective neurons. Top: Raster plot time-locked to stimulus onset with spike waveform with shaded standard deviation above. Bottom: Corresponding firing rates. Asterisks indicate statistical significance; Left: $p < 0.001$ (Generalized linear model with Gamma link); Right: $p = 0.036$. **c** Top: schematic representation of a decoder based on weighted averages of neuronal firing rates. Bottom: Corresponding decoding performance for different confidence levels. Horizontal lines show times of significant performance ($p < 0.05$, one-sided permutation tests to correct for multiple comparisons). **d** ECoG grid with beta coefficients for detection x confidence. Non-significant electrodes are in black. **e** Average ECoG amplitudes, aligned to stimulus onset from the electrode next to the microelectrode array. Asterisks indicate statistical significance ($p < 0.001$; linear model). **f** The EEG amplitude time-locked to stimulus onset and averaged over 18 healthy controls for a cluster of significant electrodes depicted (dashed contour) on the topographic map on the right. The colormap corresponds to the beta coefficient for the interaction effect between detection and confidence. Asterisks indicate statistical significance ($p < 0.001$; linear mixed model). In all panels, misses are depicted in red, hits in blue, correct rejections in green, shaded areas represent 95% CI, black horizontal bars represent the analysis window for statistics.

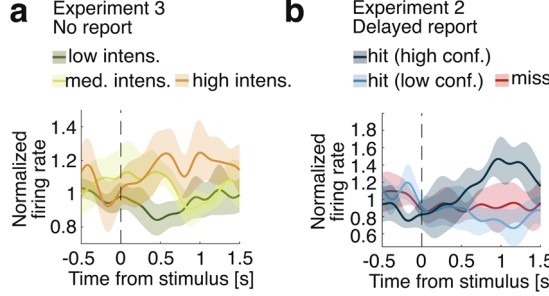

**Fig. 3 Average firing rates of responsive neurons.** Firing rates were normalized using a 0.3 s pre-stimulus baseline. **a** Normalized firing rate for three bins of stimulus intensity (low intensity in dark green, medium intensity in light green, high intensity in orange), averaged across intensity-selective neurons ($N = 14$). In Experiment 3, the participant provided no detection or confidence report and was let to mind wander. **b** Normalized firing rate for high and low confidence for hits (blue) and for misses (red), averaged across all detection- and confidence- selective neurons with higher firing rates for hits ($N = 10$). In Experiment 2, the participant waited at least 1 s before reporting detection and confidence vocally.

neuron showing a significant correlation between spike count and confidence in misses. The same pattern was observed in the population of 86 neurons, as higher correlation coefficients between spike counts and confidence were positively related with choice probabilities for hits but not for misses (Supplementary Fig. 4a–b).

To better characterize how neuronal population activity relates to detection and confidence, we trained decoders on the firing rate of all neurons and evaluated them out-of-sample. We decoded hits from misses better than chance for both high confidence (Fig. 2c; max. area under the curve (AUC): 0.88, 1.16 s after stimulus onset) and low confidence (max. AUC: 0.63 accuracy at 0.77 s). This indicates that although low confidence hits and misses were indistinguishable based on individual neurons they could be discriminated at the population-level, which confirms that our results were not driven by high-confidence trials only. The output of the best decoder (at 1.13 s) correlated with confidence for hits ($R = 0.58$; $p = 0.001$, permutation test) but not for misses ($R = -0.15$; $p = 0.13$), confirming that the neuronal signal driving detection also explains confidence for detected stimuli. Finally, we also found interaction effects between detection and confidence in ECoG electrodes

surrounding the microelectrode array (Fig. 2d–e) and in electroencephalography (EEG) signals from 18 healthy volunteers in Experiment 4 (Fig. 2f), consistent with previous EEG studies[19,38]. Together, results from Experiments 1 and 2 show that PPC neurons exhibit evidence accumulation behavior and encode detection and confidence reports irrespective of motor actions and report effector (i.e., keypress in Experiment 1, voice in Experiment 2). Of note, the latency of the evidence accumulation process we uncovered in Experiment 2 is qualitatively compatible with the distribution of RTs measured in Experiment 1, which suggests that perceptual consciousness occurs with a delay of up to 1 s following weak vibrotactile stimulation.

**Experiment 3: no-report paradigm.** While the use of delayed responses in Experiment 2 ensured that the comparison of hits and misses was not contaminated by motor confounds, a possibility remains that the neurons we uncovered were involved in task execution rather than subjective experience per se. To distinguish neuronal activity associated with subjective experience from activity associated with subjective report[8,39,40], in Experiment 3 we let the participant mind wander while they were exposed to stimuli ranging between 0.5 and 5 times the perceptual-threshold intensity. We reasoned that neuronal activity would still encode the intensity even for task-irrelevant stimuli if evidence accumulation determines conscious perception beyond mere reports. While no behavioral task was enforced, we found that the activity of 14/96 neurons increased with increasing stimulus intensity (14.58%, $p = 0.008$; Fig. 3a). The fact that stimulus intensity was represented at the single-neuron level although the participant was not engaged in the task argues against the possibility that our previous results in Experiments 1–2 reflected activity related to reporting rather than perceptual processing leading to consciousness[8,39,40].

**Experiment 4: computational model of detection and confidence.** Informed by these human single-neuron data from Experiments 1–3, we sought to generalize our decisional account of perceptual consciousness and monitoring by identifying evidence accumulation mechanisms underlying detection and confidence in EEG data (18 healthy volunteers, task similar to Experiment 2). Participants behaved similarly to the aforementioned patient, with a balanced number of hits and misses (Supplementary notes) and EEG responses also showed an interaction effect between detection and confidence (Fig. 2f). We developed an evidence accumulation model to fit the behavioral and EEG data, assuming that participants attempted to detect the stimulus by continuously accumulating evidence during a 3 s stimulation window (from trial onset until the response cue). To model the time uncertainty in our task (participants did not know when a stimulus could be applied), we assumed that participants started accumulating evidence before the stimulus onset[41]. This was modeled as a null drift rate across time except for a short-lasting boost triggered by the stimulus. A stimulus was perceived if the simulated evidence accumulation (EA) process reached a bound[16] at any time during the stimulus window (Fig. 4a), compatible with all-or-none views of consciousness[42].

Confidence was readout from the difference between accumulated evidence and the decision threshold[32,33], with a sign inversion for misses and correct rejections, since the closer the evidence to the bound, the lower the confidence (Fig. 4b). Importantly, we sampled confidence when evidence reached a maximum across the stimulation window, which allowed implementing a confidence readout for misses and correct rejections, for which no decision threshold is crossed. To fit the model parameters to the data, we considered the shape of the electrophysiological signature for hits and misses as a neural

correlate of evidence accumulation[19,43,44], defined by the weighted average of all EEG electrodes that maximally discriminated hits from misses. We first fitted the parameters of a detection model to these electrophysiological responses (Fig. 4c, Supplementary Fig. 5) as well as to hit and false-alarm rates (Fig. 4d; Supplementary Fig. 6). We then fitted two additional parameters for confidence bias and sensitivity to observed confidence distributions. The resulting model fitted the confidence ratings well (average R across participants $0.80 \pm 0.03$ for hits, $0.83 \pm 0.03$ for misses, $0.82 \pm 0.04$ for correct rejections, and $0.33 \pm 0.10$ for false alarms; Fig. 4e–f, Supplementary Fig. 7), suggesting that evidence accumulation is a plausible mechanism underlying both perceptual consciousness and monitoring. The data and the model were still consistent when stratifying per confidence level. Metacognitive sensitivity predicted by our model and observed in the data were correlated for both "yes" responses ($R = 0.66$, $p = 0.000$, permutation test) and "no" responses ($R = 0.62$, $p = 0.007$; Fig. 4g), showing that our model also successfully predicted metacognitive performance. Finally, an alternative model assuming that confidence for detected stimuli is sampled at a fixed latency after crossing the decision threshold and confidence for undetected stimuli is sampled from a random distribution led to a worse fit of the data (BIC $= 229.31 \pm 43.19$ compared to BIC $= 178.82 \pm 38.86$ for the maximal evidence model; $z = -2.33$, $p = 0.020$; Fig. 4h).

To support the generalizability of our model, we verified that it could reproduce qualitatively similar evidence accumulation traces as observed in neuronal data from Experiment 1 (Supplementary Fig. 8). We also assessed whether we could decode confidence for misses from single-neuron data in Experiment 2 using a decoder defining confidence as the maximum of accumulated evidence. We took the best decoder for hits and misses (trained on stimulus-locked data) and applied it out-of-sample across the stimulation window (i.e., cue-locked). We decoded confidence for hits ($R = 0.43$, $p = 0.001$, permutation test) and confidence for misses ($R = -0.26$, $p = 0.037$), which the aforementioned stimulus-locked decoder (Fig. 2c) could not achieve. The time corresponding to the decoded maximal evidence correlated with stimulus onset for hits ($R = 0.23$, $p = 0.046$) but not for misses ($R = 0.16$, $p = 0.12$), suggesting that evidence for confidence in misses was not sampled synchronously with the stimulus, thereby verifying the plausibility of the maximal evidence decoder on our patient's single-neuron data.

## Discussion

We propose a mechanism of evidence accumulation to explain the behavioral and neural markers of perceptual consciousness and monitoring. We show that tactile detection relates to an increase of the firing rate of single neurons and ECoG amplitude in the posterior parietal cortex of a human participant, as well as an increased scalp EEG response recorded in a group of healthy participants. In both cases, the amplitude of the corresponding neural response was dependent on the confidence in hits. This increase in neural response as well as in the detection reports were well described by a computational model indicating that a plausible mechanism underlying the building of confidence in both the presence and absence of a stimulus is for the brain to take the maximal evidence accumulated over time. Using a combination of delayed response and no-report paradigms, we ruled out the possibility that these effects stemmed from motor actions or task demand.

We had the opportunity to collect data from individual neurons in the human PPC, at the junction between the postcentral and intraparietal sulcus in the superior parietal lobule. The PPC has been associated with a multitude of functions linking

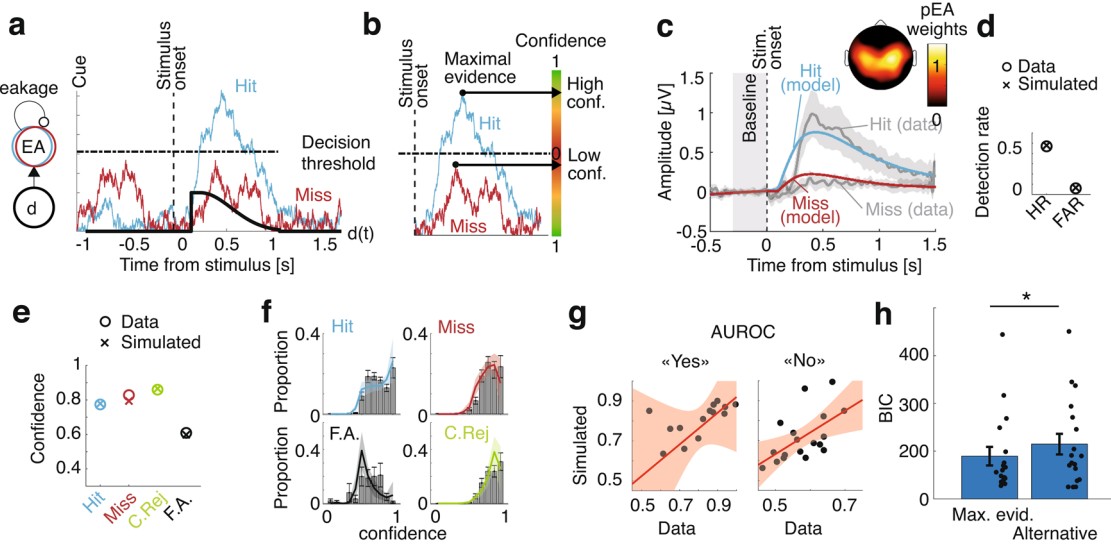

**Fig. 4 Computational model based on evidence accumulation. a** Simulation: Time-varying drift rate (d; thick black trace) had a short-lasting boost after a non-decision time following stimulus onset (dashed vertical line). Example evidence accumulation for one trial (EA; blue trace for a hit, red trace for a miss) rises sharply after the drift boost and is attracted back to zero due to leakage. A stimulus is considered as perceived (hit) if EA reaches a decision threshold (horizontal line), and as non-perceived (miss) if not. **b** The maximum of accumulated evidence with respect to the decision threshold is used as a confidence readout. In the example in **a**, the blue trace results in high confidence in a hit (large distance from the bound) and the red trace results in low confidence in a miss (small distance from the bound). **c** Model fit of the proxy to EA (pEA) locked on stimulus onset for hits (blue trace) and misses (red trace). The corresponding observed EEG data for hits (upper line) and misses (lower line) is shown in grey. Average scalp topography of normalized pEA weights is shown above. Shaded areas represent 95%-CI. **d** Hit rate (HR) and false-alarm rate (FAR). Datapoints are represented as 'o' and model simulations as 'x'. **e** Average confidence for hits (cyan), misses (red), correct rejections (green) and false alarms (black). **f** Model fits of the confidence distributions. Histograms show confidence distributions with 95%-CI whiskers. Colored traces show model simulations. All shaded areas represent 95%-CI. The y axis represents the proportion of trials. **g** Area under the receiver operating characteristic curve (AUROC) correlation between observed data (horizontal axis) and simulated data (vertical axis) for "yes" responses (hits and false alarms; left) and "no" responses (correct rejections and misses; right). Regression line is shown in red with shaded areas representing 95%-CI. **h** Model comparison in terms of Bayesian information criterion (BIC) between the maximal evidence model and the alternative model. Whiskers represent 95%-CI and asterisk indicates statistical significance (p = 0.0020, N = 18 participants, Wilcoxon signed rank).

perception to planning and action[45] and receives multisensory inputs including those from the primary somatosensory cortex[46]. In Experiment 1, we found individual neurons in the PPC with higher firing rates following detected stimuli. We argue that these neurons are responsible for evidence accumulation based on the following three findings. Firstly, in Experiment 1 we found neurons with a gradual buildup of firing rates for hits prior to response times[47]. Secondly, in Experiment 3, we observed an increase in the firing rates for increasing intensities of (task-irrelevant) stimuli. Based on these two hallmarks of evidence accumulation, we argue that our results consist in the first single-neuron account of perceptual evidence accumulation in a human subject capable of subjective reports. Indeed, although electrophysiological correlates of evidence accumulation have been found in various regions of non-human primate brains, including the frontal cortex or subcortical structures[48–50], the arguably most common region studied in relation with neural accumulation of perceptual evidence is the lateral intraparietal (LIP) area of the PPC[13,14]. However, whether perceptual evidence accumulation neurons such as those reported in non-human primate studies could support conscious reports is unclear, as subjective experience cannot be measured explicitly in non-human species, and because such neurons were—to our knowledge—not reported in humans yet.

In Experiment 2, we asked the participant again to detect stimuli and found neurons similar to those in Experiment 1 with higher firing rates after stimuli reported as perceived. The finding of detection-selective neurons when responses were provided by key press (Experiment 1) or orally (Experiment 2) suggests that

the mechanism of evidence accumulation we propose is response-invariant. Importantly, we asked the participant to report the confidence they had in their responses, and found that the change in firing rates for detected stimuli was modulated by confidence, showing that confidence relates to the strength of single neuron's responses to detected stimuli. This mechanistic overlap, which—to our knowledge—was not yet shown in humans capable of reporting subjective confidence was confirmed at the neuronal population level: multivariate decoders trained to discriminate hits vs. misses allowed us to decode confidence for hits when time-locking to the stimulus onset and for both hits and misses when locking on the onset of maximal evidence. This result implies the existence of neurons aggregating the output of evidence accumulating neurons, which remain to be described.

Because microelectrode implants in parietal regions are rare in humans, we sought to generalize our findings by recording behavioral and neural data in a group of healthy volunteers in Experiment 4. Behavioral results revealed highly similar patterns between the two samples, indicating that detection and confidence reports in Experiments 1–2 were not impacted by the clinical condition. Analyzing the amplitude of ECoG data and EEG data from Experiment 4, we could generalize our single-neuron findings at a larger scale and arguably distinct electrophysiological processes. EEG data showed a similar dependence of detection-related activity on confidence for hits, similar to previous work in the visual domain using a different awareness scale[19] and a discrimination task[51,52]. Of note, both neural responses recorded at the intracranial (single-neuron, ECoG; Experiment 1–3) and scalp levels (Experiment 4) were observed at

a rather late latency following stimulus onset (>200 ms), suggesting that these responses were not related to early somatosensory perceptual processes. We then reasoned that since similar increases in neural activity are assumed to reflect accumulation of evidence[43], an evidence accumulation model should predict both behavioral results (hit rate, false-alarm rate and confidence for hits, misses, correct rejections and false-alarms) and corresponding neural responses. Using neural data to fit the model instead of response times allowed us to fit a leakage parameter[53] and to compensate for the fact that, in a detection task, response times are unavailable for undetected stimuli. The model derives confidence as the distance between the maximal evidence accumulated over time and the decision bound. This flexible readout provides a major advantage when computing confidence in the absence of stimulus detection (i.e., misses) as well as in the absence of a stimulus (i.e., correct rejections and false-alarms). This could not be achieved with previous models using decision-locked confidence readouts in discrimination tasks[32–34]. An alternative model with a fixed timing readout and a random confidence for unperceived stimuli performed significantly worse. It remains unknown however, how this readout is implemented in the brain, as we had limited anatomical coverage in this individual patient, and did not find parietal neurons encoding confidence irrespective of detection response.

To achieve this confidence readout, we relied on two modeling assumptions: that evidence accumulation (i) continues after reaching the bound and (ii) leaks information. Indeed, LIP neurons documented in non-human primate studies show a sharp drop in firing rate immediately after the decision, consistent with an absorbing bound, implying that evidence accumulation stops after a decision. Such a mechanism would lead to constant maximal evidence for "yes" responses, and therefore require a second accumulator towards the opposite choice to derive graded confidence ratings[54]. This second accumulator may be represented by the neurons we found showing a higher increase in firing rate for misses compared to hits, raising also the intriguing possibility that misses are encoded actively. This extends a study on non-human primates which found prefrontal neurons also coding for misses but rather in the delay period following the stimulus[22]. However, more work is needed to show how such neurons could contribute to confidence in detection tasks.

Contrarily to LIP neurons, the firing rate of the neurons we found in experiment 1 continued to increase shortly after the response. Although such post-decisional evidence accumulation is inconsistent with LIP data, it is supported by strong computational[55] and electrophysiological[56] evidence and extensive work in humans rely on post-decisional accounts of confidence ratings[33,34,51].

In our study, leakage also allowed us to reproduce the decrease in EEG or firing rate we observed, subsequent to the stimulus onset or response. This is again inconsistent with studies using random-dots kinematograms, showing evidence for perfect accumulation both at the behavioral[57] and neural levels[58]. In such discrimination tasks however, stimulus onset is a clearly defined event and perfect accumulation between stimulus onset and response can be considered as optimal. The picture is very different in detection tasks with unpredictable stimulus onsets, such as in this study. Indeed, when the drift rate is zero (i.e., before stimulus onset), a perfect accumulator with a reflexive bound at zero would integrate noise. In such cases, leakage[59] has been shown to occur[60] and could represent an efficient mechanism to keep the accumulator close to zero and avoid integrating noise resulting in false alarms.

Apart from being useful for first-order detection responses, a confidence model based on leakage has been shown to be consistent with empirical data[53]. Nonetheless, the question of whether evidence accumulation is perfect or leaky remains open. Furthermore, a recent study comparing models based on signal detection theory showed that the model that best fit observed data involves a second-order "metacognitive" noise to the decisional evidence[61]. Our model implicitly implements this metacognitive noise through the influence of leakage on post-decisional evidence readouts. Indeed, in participant with strong leakage, accumulated evidence rises and decays fast, leading to low metacognitive noise. On the contrary, in participants with little leakage, once no more informative evidence is accumulated, the level of evidence accumulation tends to oscillate around the reached maximum, leading to higher metacognitive noise. A-posteriori analyses showed that the leakage parameter correlated with metacognitive sensitivity (Supplementary notes). Our model thus provides a simple mechanism supporting metacognition, including subliminal stimuli, explains neural responses and was verified at the single neuron and EEG level, as we were able to decode confidence ratings above chance using this procedure.

Our results posit that stimulus detectability involves the accumulation of sampled evidence towards a decision bound, as previously discussed for discrimination tasks[16] as well as several cognitive functions. Further research will be needed to assess whether this mechanism generalizes to other perceptual domains beyond vibrotactile stimuli. Of note, the use of a detection task is compatible with a contrastive study of consciousness[6], as opposed to two-alternative forced choice discrimination tasks for which confidence ratings are well characterized, but which do not offer a direct contrast between perceived and unperceived stimuli. The mechanism we propose implies that perceptual consciousness is an all-or-none process involving a threshold, compatible with several theoretical accounts[62]. One could speculate that evidence accumulates in the PPC until crossing a threshold, which in turns triggers the coalescence of multiple encapsulated networks into a single network responsible for broadcasting neural signals throughout the brain (i.e., ignition according to the global workspace theory of consciousness[11]). To determine whether PPC serves as an ignition trigger will require electrophysiological recordings with broader coverage. Furthermore, conceptual work will be needed to assess to what extent evidence accumulation is necessary if not sufficient for consciousness[63,64], and how it fits within other theoretical frameworks such as higher order thought theories[26] or recurrent processing theories of consciousness[65].

Recently, the use of classical contrastive approaches to delineate the neural correlates of consciousness has been criticized, on the basis that it may be confounding the cognitive and neural mechanisms associated with phenomenal experience per se, and those associated with reporting phenomenal experience[8]. Some authors have proposed the use of "no-report paradigms", in which perceptual experience is not inferred from participants' responses, but from neural or peripheral signals while participants are passively exposed to stimuli[35,40,66,67]. Importantly, we found a population of neurons encoding perceptual evidence through a putative evidence accumulation process in Experiment 3, in which the participant was passively exposed to the stimuli similar to such no-report paradigms. Although these effects were weaker than the ones found in Experiment 1–2, they indicate that evidence accumulation operated by a neuronal population in the posterior parietal cortex is involved in conscious perception, even when the stimuli are task-irrelevant. In addition, the mechanistic overlap between detection and confidence we report cannot be due to similar motor responses associated with detection and confidence reports in Experiment 2, as those were collected separately, seconds after the end of the stimulation window on which our analysis was based in the delayed detection task.

To conclude, our results posit that both detection and confidence for near-threshold stimuli involve the accumulation of evidence towards a criterion orchestrated by the PPC. We argue that this neuronal mechanism involving a decision bound may serve as a trigger for the neural ignition underlying perceptual consciousness[12,68] and explains how contents remaining inaccessible to perceptual consciousness may still be subject to perceptual monitoring[25,69]. Our behavioral, neural, and modeling results clarify how perceptual consciousness and perceptual monitoring are intertwined mechanistically.

## Methods

**Participants**. In experiments 1–3, the participant was a young right-handed adult suffering from drug-resistant epilepsy due to a focal cortical dysplasia in the left central sulcus. As part of the clinical management of their condition, the patient received a 4x6 ECoG grid covering the left premotor, motor, sensory and posterior parietal cortices. The patient accepted to participate in a clinical trial on neuronal recordings during invasive epilepsy monitoring at the Geneva University Hospitals (IN-MAP; NCT02932839) and a Utah microelectrode array was additionally implanted in the left posterior parietal cortex. Data collection for this clinical trial was completed in January 2021: results are being analyzed and will be the subject of a later scientific communication. The present work is independent of the clinical trial. A data sharing agreement was established between the sponsor, the clinical institution, and the recipient third party. The main terms of this contract were as follows: the data remain owned jointly by the sponsor and the clinical institution, and are provided to the recipient solely for the purpose of addressing a specific field of research. None of the pre-specified outcomes of the clinical trial are mentioned in the present manuscript. The patient provided informed written consent for the present study. The present study was approved by the Commission Cantonale d'Ethique de la Recherche de la République et Canton de Genève (2016-01856). Except for ibuprofen, paracetamol and esomeprazole, the patient was not under the influence of any medication (i.e., no antiepileptic drug). The patient rated their pain at 1 out of 10 on a visual analog scale, except for Experiment 3, for which they rated their pain at 2 out of 10, reporting a slight pain around the left orbit. No neurological or cognitive abnormality, or anxiety were reported by the clinical team and the patient's mood was stable. To ensure adequate power, the sample size of the EEG experiment was pre-determined on the basis of prior experimental works assessing perceptual decisions under temporal uncertainty[41]. Eighteen healthy participants (seven females; age: 25.2 years, SD = 4.1) took part in Experiment 4 for a monetary compensation. All participants reported being right-handed, had normal hearing and normal or corrected-to-normal vision, and no psychiatric or neurological history. Two participants were excluded due to excessive artifacts. Participants gave written informed consent prior to participating and all experimental procedures were approved by the Commission Cantonale d'Ethique de la Recherche de la République et Canton de Genève (2015-00092 15-273).

**Procedure**. Experimental paradigms were written in Matlab (Mathworks) using the Psychophysics toolbox[70–72] (Table 1). In all experiments, stimuli were applied on the lateral palmar side of the right wrist using a MMC3 Haptuator vibrotactile device from TactileLabs Inc. (Montréal, Canada) driven by a 230 Hz sinusoid audio signal lasting 100 ms. Experiments started by a simple estimation of the individual detection threshold. The tactile stimulus was applied with decreasing intensity with steps corresponding to 2% of the initial intensity until the participant reported not feeling it anymore three times in a row. We then repeated the same procedure but with increasing intensity and until the participant reported feeling the vibration three times in a row. The perceptual threshold was estimated to be the average between the two thresholds found using this procedure. This approximation was then used as a seed value for an adaptive staircase during the main experiments (see below). Experiments 1–3 were performed on different days at the patient's bedside.

*Experiment 1*. Stimuli ($N = 150$/session; two sessions) were applied in a pseudo-random way with an inter-stimulus interval of two seconds plus an exponentially distributed time (mean: 2 s). The participant was provided with a keypad and asked to press a key every time they felt a stimulus. Answers provided during the two seconds following a stimulus were considered as hits. Only one keypress occurred out of this two second post-stimulus window.

*Experiment 2*. An auditory cue signaled the start of the two seconds stimulus window ($N = 150$; one session) during which the stimuli could be applied at any time (uniform distribution) in 80% of trials (the remaining 20% served as catch trials, unbeknownst to the participant). Stimulus window was followed by a one second delay to ensure that stimulus-locked activity was not contaminated by the detection response. After this delay, a second auditory cue probed the participant for their detection response ("yes" or "no"), followed by a three levels confidence rating (1: "unsure", 2: "somewhat sure", 3: "very sure"). Detection and confidence ratings were provided vocally and registered by the experimenter.

*Experiment 3*. Stimuli ($N = 220$; one session) were applied in a pseudo-random way with an inter-stimulus interval of two seconds plus an exponentially distributed time (mean: 2 s) with a random amplitude sampled from 11 intensities ranging from zero to five times the participant's perceptual threshold. The participant was not given any instructions and was left free to mind wander during the experiment.

*Experiment 4*. Participants sat in front of a computer screen. A white fixation cross appeared in the middle of the screen for 2 s. From the moment the fixation cross turned green, participants were told that a tactile stimulus could be applied at any moment during the next 2 s. During this period, stimulus onset was uniformly distributed in 80% of trials, the 20% remaining trials served as catch trials, as in Experiment 2. In all trials, 1 second after the green cross disappeared, participants were prompted to answer with the keyboard whether they felt the stimulus or not. Following a 500 ms stimulus onset asynchrony, participants were asked to report the confidence in their first order response by moving a slider on a visual analog scale with marks at 0 (certainty that the first-order response was erroneous), 0.5 (unsure about the first-order response) and 1.0 (certainty that the first-order response was correct). Detection and confidence reports were provided with the left (non-stimulated) hand, using different keys. The total experiment included 500 trials divided in 10 blocks and lasted about 2 h.

**Electrophysiological data acquisition**. A 96-channel silicon-based microelectrode array ("Utah array"; Blackrock Microsystems, Salt Lake City, USA) was implanted in the posterior parietal cortex, immediately posterior to the postcentral sulcus and the hand representation of sensorimotor cortex (Fig. 1A). The location was confirmed through post hoc electrode localization (Fig. 1G), performed through a coregistration of a preoperative MRI structural T1 scan and a postoperative CT scan using the iELVIS toolbox[73]. The data from each of the 96 channels was amplified and sampled at 30 KHz for offline analysis (NeuroPort system, Blackrock Microsystems LLC, Salt Lake City, USA). In addition, a 24 electrode ECoG grid (Ad-Tech Medical) covered the left hemisphere from the premotor cortex to the superior parietal lobule (Figs. 1G, 2D). The electrodes had a 4 mm diameter with 2.3 mm exposed corresponding to an area of 4.15 mm². The data were amplified and sampled at 2048 Hz (Brain Quick LTM, Micromed, Treviso, Italy). In Experiment 4, electroencephalographic data were acquired from 62 active electrodes (10–20 montage) using a WaveGuard EEG cap and amplifier (ANTNeuro, Hengelo, The Netherlands) and digitized at a sampling rate of 1024 Hz. Horizontal and vertical electrooculography (EOG) was derived using bipolar referenced electrodes placed around participants' eyes. The audio signal driving the vibrotactile actuator was recorded as an extra channel to precisely realign data to stimulus onset.

**Statistical analysis**. We used non-parametric tests (permutations and Wilcoxon signed-rank tests).

**Invasive electrophysiological data processing and univariate analysis**. The raw signal from the microelectrode array was bandpass filtered between 300 and 3000 Hz for spike sorting. Trials with epileptic activity or other artifacts were removed from further analysis following visual inspection of ECoG data. Spikes were extracted and sorted using the semi-automatic template matching 'Osort' algorithm[74]. Standard quality metrics were computed for each putative single unit in order to assess their quality (Supplementary Fig. 1). We computed the firing rate every 1 ms with a 100 ms standard deviation Gaussian sliding window.

In Experiment 1, a neuron was considered detection-selective when a significant (two-sided test) effect of detection was found on the number of spikes during a time window between 0.5 and 1.5 s after stimulus onset using a Wilcoxon signed-rank test. For the latency analysis, we fitted a regression line to the firing rate between 300 ms and the RT, for each trial independently. A neuron was considered RT-selective if the Spearman correlation between the slope of the regression line and the RT was significant. We also computed neuronal choice probabilities for detection using the area under the receiver operating characteristic's (AUROC) curve based on the spike counts during the 0.5–1.5 s window after stimulus onset, normalized to the $[-1,1]$ interval: $CP = 2*AUROC-1$. To ensure that there was no overfitting and that our results were not driven by outliers, we used a non-parametric permutation test to assess whether the number of selective neurons was significantly above chance; we repeatedly ($N=1000$) applied the same tests on shuffled data and counted the number of selective neurons. We defined the p-value as the proportion of times that the number of selective neurons for shuffled data was higher than the number of selective neurons found in the data[37,75]. When no selective neuron was found in the shuffled data, we set $p = 1/N = 0.001$.

In Experiment 2, we also used a generalized linear model (GLM) with a Poisson distribution[37,76] to regress the number of spikes during a time window between 0.5 and 1.5 seconds after stimulus onset. We fitted a model with three beta regressors: #spikes ~β0 + β1*detection + β2*confidence + β3*detection*confidence. We only interpreted main effects (β1, β2) in the absence of interactions (β3). If β3 was significant (two-sided test), we considered the neuron as detection- and confidence-selective. If β1 was significant but β3 was not, we considered the neuron only detection-selective (idem for confidence-selective). We applied the same permutation test as for Experiment 1. In Experiment 3, we fitted a Poisson GLM

with one beta regressors: #spikes ~β0 + β1*intensity to find intensity-selective neurons showing an increased firing rate with increasing stimulus intensity and applied the same permutation test as for Experiment 1 and 2.

For ECoG analyses, we re-referenced the channels to a common average and applied a lowpass filter (Hamming window with a cutoff frequency of 40 Hz). Trials with epileptic activity or other artifacts were removed from further analysis following visual inspection. We used linear models (LM) for statistics using the same regressors as for spike counts (see above). For display purposes only, we additionally smoothed the data with a 200 ms Savitzky–Golay filter[77].

**Multivariate decoding (Experiment 2)**. We fed single neurons firing rates sampled every 10 ms into linear discriminant decoders with a L2 regularization factor of 0.8 (similar results were obtained with different regularization factors). These decoders separated the space of input features with a linear hyperplane that best discriminates hits and misses. The decoders predict a hit when the distance of a sample to the separating hyperplane is higher than zero and a miss otherwise. To avoid overfitting, we separated our data in 10 cross-validation folds so that for each fold we trained decoders on the 90% of the data and tested them on the 10% remaining (i.e., out-of-sample). The distance to the separating hyperplane was also used to compute the area under the receiver operating characteristic's curve (AUROC) for out-of-sample data at each time point. We used permutation tests to assess whether similar AUROC could have been obtained by chance while correcting for multiple comparisons across time points. We selected contiguous clusters of time points when AUROC was higher than 0.6 and then computed the proportion of similar or bigger clusters obtained with shuffled labels. For each permutation, we computed AUROC clusters over the whole set of time points to keep the autocorrelation structure in the shuffled data.

Confidence was decoded at the time point corresponding to the highest AUROC. Decoded confidence was defined as the absolute distance of a sample to the separating hyperplane. We assessed confidence decoding by correlating (Spearman) decoded confidence with observed confidence and assessing significance using permutation tests (one-sided, as we could reasonably expect positive correlations for hits and negative correlations for misses). As this procedure was carried out on out-of-sample data, our results were not affected by overfitting. Finally, for maximal evidence decoding, we used the same decoder to decode firing rates between 0.8 and 3 s post cue. We excluded the first 0.8 s due to some neurons showing post-cue activity. We then took the maximum of the decoder over that time window, correlated it with confidence observed in the data and assessed significance using permutation tests (one-sided). Again, this procedure was carried out on out-of-sample data so our results cannot be affected by overfitting.

**Scalp EEG data preprocessing (Experiment 4)**. All channels were highpass filtered using a Hamming window with a cutoff frequency of 0.1 Hz. We defined an epoch as the 3 seconds of data centered around the event corresponding to the vibrotactile stimuli recorded using an auxiliary channel. EEG and EOG data were then lowpass filtered using a Hamming window with a cutoff frequency of 40 Hz and visually inspected to remove trials and channels containing artifacts. We computed the independent component analysis (ICA)[78] on a copy of the EEG epochs that were highpass filtered at 1Hz. The number of independent components (ICs) computed corresponded to 99% of the variance, which resulted in 14.78 ± 1.27 ICs per subject. We used SASICA[79] with default parameters to automatically select ICs for rejection and visually inspected all components scheduled for rejection before actually rejecting them. IC weights kept were then back-projected to the original EEG epochs. Any channel rejected prior to the ICA was reinterpolated using spherical interpolation ($N = 0.67 ± 0.23$, max 3). Finally, we visually inspected all channels and rejected artifactual epochs. All pre-processing was done with the EEGLAB toolbox[80]. The final dataset comprised 464.17 ± 13.73 epochs per subject.

To assess which electrodes were detection- and/or confidence-selective, we used a linear mixed model to regress single-trial average EEG responses in the 0.5 to 1.5 time-window used for single-neuron analysis. For each electrode, we tried different random factors and kept the model with the lowest Bayesian Information Criterion. P-values were Bonferroni corrected for multiple comparisons. To compute the electrophysiological correlate of evidence accumulation, we sought to find the best weighting of EEG electrodes in terms of discriminability between hits and misses. For this, we trained decoders of hits versus misses between −0.5 and 1.5 s from stimulus onset with 20 ms steps in a 10-fold cross-validation scheme. We repeated this procedure 10 times and searched for the time point with highest average discriminability. We then retrained one decoder using the EEG at this time point and used the weights of this decoder to construct one single value at every time point, representing a proxy to the amount of evidence for hits. We baselined this proxy signal using the 300 ms pre-stimulus.

**Evidence accumulation model**. To test whether the observed electrophysiological correlate of conscious detection could index evidence accumulation, we used an evidence accumulating EA(t) process consisting of a diffusion process plus a drift with a leakage parameter driving the accumulated evidence back to zero[81]. The model thus integrated a time-varying drift rate d(t) with a leakage factor λ plus

additive white noise W(t) with a fixed standard deviation of $\sigma = 0.1$ (Eq. 1). The evidence accumulation process EA(t) was bounded by zero to be more biologically plausible (since firing rates are positive).

$$EA(t+1) = \max([1 - \lambda * dt] * EA(t) + d(t) + \sigma W(t), 0) \tag{1}$$

To model temporal uncertainty, the accumulation process started at the beginning of the stimulus window[41] with a drift of zero and ended 3 s later, as in our experimental paradigm. On stimulus onset, and after a non-decision time (ndt), the drift rate d(t) rose to a level γ for 100 ms (stimulus duration) and then decayed exponentially with a factor k (Eq. 2). We used the same distribution of stimulus onsets as in the data (from 0 to 2 s after the cue). To model variability in the drift rate for a detection task[82], γ was sampled from a half-normal distribution with mean $\gamma\_\mu$ and standard deviation $\gamma\_\sigma$ (i.e., absolute value of a normal distribution). This allowed us to have variability in the drift rate while keeping it positive.

$$d(t) = \begin{cases} 0, & \text{if } t < s + ndt \\ 1, & \text{if } (s + ndt) \le t \le (s + ndt + 0.1) \\ \gamma e^{[t-s-ndt-0.1/k]^2}, & \text{if } t > (s + ndt + 0.1) \end{cases} \tag{2}$$

With s the onset of the stimulus. Stimuli were considered to be detected if the accumulated evidence reached a decision bound θ.

Confidence c(t) was simulated as the accumulated evidence EA(t), scaled by a factor α and shifted by a factor β (confidence bias), inverted if the decision bound was not reached (invert (x) = 1 − x) before being saturated to the 0–1 interval by a sigmoidal function[32] (Eq. 3). The time at which confidence c(t) was read-out corresponded to the maximum of EA(t) over the 3 s stimulation time window. Of note, we expressed the confidence readout in percentage of the decision bound. This scaling did not affect simulated confidence but normalized the readout across participants and helped restrain the grid search for good initial parameters.

$$c(t) = (e^{\alpha * EA(t)+\beta})/(e^{\alpha * EA(t)+\beta} + 1) \tag{3}$$

**Model fitting**. We used a two-stage fitting procedure: We first fitted the parameters of the decisional process to detection responses and to the pEA described above and then fitted a second set of parameters to predict confidence ratings.

In the first stage, we simulated (N=500) trials of EA(t) along with the corresponding detection responses. The objective function of the optimization procedure was based on the likelihood of the model with respect to the hit rate and false-alarm rate observed in the data and the shape of the average of pEA(t) for hits and misses. Since our model is agnostic to the scale of pEA(t) and EA(t), we scaled them both by their average over time and realizations (trials or simulations). The log-likelihood thus corresponded to the log of the normal probability of observing such a hit rate and false-alarm rate and the log of the normal probability of observing such an electrophysiological response for hits and misses. The standard deviation of the observation noise was set to 0.02 for hit rate and false-alarm rate and 0.2 for EA(t) and pEA(t). We used a Nelder–Mead simplex optimization with 756 different initial parameters sampling a broad range of values for $\gamma\_\mu$, $\gamma\_\sigma$, λ, and k. For each such iteration, we first did a grid search on ndt and θ to find plausible starting values. We kept the parameters corresponding to the model with the best likelihood. To ensure a good fit, we did a final fitting with N=10'000 simulated trials.

In the second stage, we also simulated (N=500) trials of EA(t) along with the corresponding detection responses and used these to simulate confidence ratings. We used a Kolmogorov–Smirnov test for the log-likelihood of confidence for hits, misses and correct rejections. We used a Nelder–Mead simplex optimization with 66 different initial parameters sampling a broad range of values for α and β. We kept the parameters corresponding to the model with the best likelihood. To ensure a good fit, we did a final fitting with $N = 10,000$ simulated trials.

**Metacognitive sensitivity**. We evaluated metacognitive sensitivity or how well confidence predicted task performance[23]. For this, we represented the posterior probability of confidence ratings knowing the correctness of the detection report using the area under the receiver operating characteristic (AUROC) curve, computed independently for "yes" responses (hits and false alarms) and "no" responses (correct rejections and misses).

**Alternative model**. We compared our maximal evidence model with an alternative model which assumed that confidencce was readout at a fixed timing (t_RO) post decision for perceived stimuli. For unperceived stimuli (i.e., with no decision), this alternative model assumed that participants reported random confidence estimates, based on Gaussian noise with mean Ω and unit standard deviation. As in the maximal evidence model, confidence corresponded to evidence scaled by a factor α and shifted by a factor β (confidence bias) before being saturated to the 0–1 interval by a sigmoidal function. The model thus comprised four parameters: α, β, t_RO and Ω which were fitted with the same procedure as for the maximal evidence model, except that the grid search for optimal initial parameters was extended to include two initial values for t_RO: 0.25 and 0.5 s. We used the Bayesian Information Criterion to compare the two models.

**Statistics and reproducibility**. Experiments were not repeated independently to assess reproducibility.

**Reporting summary**. Further information on research design is available in the Nature Research Reporting Summary linked to this article.

## Data availability

Behavioral, electrocorticographic, and electroencephalographic data that support the findings of this study have been deposited in a public repository (OpenNeuro: https://doi.org/10.18112/openneuro.ds001785.v1.1.1)[83]. Microelectrode data are available upon written request, dependent upon the establishment of a data sharing agreement between the trial's investigator, sponsor, and interested third party. Source data are provided with this paper.

## Code availability

All analyses scripts and computational models are available in a public repository (OSF: https://doi.org/10.17605/OSF.IO/YHXDB)[84].

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

## Acknowledgements

M.P. is supported by an Early Postdoc.Mobility fellowship from the Swiss National Science Foundation (P2ELP3_187974). P.M. is supported by an Ambizione fellowship from the Swiss National Science Foundation (PZ00P3_167836). M.S. is supported by grants from the Swiss National Science Foundation (163398, 80365) and Fondation Privé. O.B. is supported by the Bertarelli Foundation, the Swiss National Science Foundation, and the European Science Foundation. N.F. has received funding from the European Research Council (ERC) under the European Union's Horizon 2020 research and innovation program (Grant agreement No. 803122). We acknowledge support from the Wyss Center for Bio and Neuroengineering and the Epileptology Unit, Division of Neurology, Geneva University Hospitals. We thank Jean-Baptiste Eichenlaub, Vincent de Gardelle, Emanuela de Falco, Liad Mudrik and Adam Shai for their comments.

## Author contributions

M.P., M.T., and N.F. developed the study concept and contributed to the study design. P.M. and M.S. developed the clinical trial that generated the microelectrode array data. Data collection and data analysis were performed by M.P., P.M., F.B., M.T., W.C., S.W., and N.F. M.P. performed modeling work. M.C. and S.M. performed surgery. M.P. and N.F. drafted the paper; A.R., O.B., and N.F. provided supervision; all authors provided critical revisions and approved the final version of the paper for submission.

## Competing interests

The authors declare no competing interests.
