## [Peer Review File · Nature Communications]

Editorial Note: Parts of this Peer Review File have been redacted as indicated to maintain patient confidentiality. Some pronouns have also been altered for this purpose.

REVIEWER COMMENTS

Reviewer #1 (Remarks to the Author):

This paper reports analyses of a very rare and valuable dataset of single neuron recordings from human posterior parietal cortex during a near-threshold detection task. The authors were able to relate the same accumulation-to-bound signals observed in single neurons to EEG recordings in healthy volunteers, and show that a computational model of evidence accumulation is able to accommodate both the neural responses and the confidence data. Perhaps most impressively, this model was able to predict confidence levels in the absence of a detection response (i.e. on miss trials) in the patient data, suggesting subtle modulations of confidence in saying “no stimulus was present” depend on the status of subthreshold evidence accumulation in the parietal cortex.

This is a tour de force combining different sets of neuronal data with an elegant and simple computational model to account for variation in perceptual confidence. I have three main comments that I hope the authors will find useful, together with minor questions about the modelling approach. I have limited knowledge of single unit analysis so my review mostly focuses on the psychophysics and interpretation of the model.

1) I found the introduction somewhat skewed by interpreting detection through the lens of perceptual consciousness. I am on board with the idea that studying detection+confidence can give us a particularly sensitive window on the computations supporting visual awareness. But I found the current framing to fall short in making this case. In particular, there is a notable failure to cite classic single-unit studies of detection in macaques – for instance Cook & Maunsell (2002) *Nat Neurosci* for detection of coherent motion, de Lafuente & Romo (2005) *Nat Neurosci* for detection of vibrotactile stimuli and Merten & Nieder (2012) *PNAS* for detection of near-threshold contrast. Many of these studies I think align with the authors' findings (of a graded accumulation that varies with stimulus intensity) albeit in different brain regions. However, and perhaps this is why they weren't included here, they are notably silent on the issue of consciousness, and instead simply present their results as neural correlates of psychophysical detection curves. I think the authors do have a strong case for framing their study as novel in this context – because the recordings are in humans, and subjective confidence reports are elicited, they offer an unusual window onto whether similar evidence accumulation processes that have been discovered in monkeys might also support perceptual awareness in humans. But the way the paper is currently structured gives the reader the impression that such studies haven't yet been done at all, in animals or humans (cf. lines 48-50).

2) I found it striking that across all experiments (and also the EEG data), misses were effectively default-coded in terms of evidence accumulation. In other words, stimuli that were missed or otherwise of low intensity led to the lowest average firing rates. This makes sense in relation to a graded magnitude code for intensity. But it is notably at odds with the notion that neural populations might actively code for each response category – i.e. an active coding of both “yes” and “no” responses (see Merten & Nieder, 2012 *PNAS* for findings on the active coding of stimulus absence in macaque PFC independent of report). Did the authors discover any neural responses in PPC that actively coded for “no” responses / confident misses? Would these have been apparent in the current analysis if they had been present? If there are none, I wonder whether this is actually an informative null result – suggesting that parietal neurons provide a low-dimensional magnitude code that might then be “read out” by the kind of PFC populations identified by Merten & Nieder, to actively code for presence and absence.

3) It took me a while to figure out how the model in Figure 4 accounted for confidence in misses. As far as I understand it from the methods, miss-confidence is inverted, so that the lower the firing rate peak, the more confident I am that nothing is there. That makes sense – but it's difficult to infer from Figure 4A. Can a similar graphical interpretation be added for miss-confidence, perhaps as a separate panel? How is this defined in relation to threshold? In relation to this point, I was initially surprised to see that the correlation coefficient between miss-confidence and activity was negative (line 263), but eventually figured out this makes sense under a definition in which miss confidence is inversely related to activity. It would be useful to help the reader through this logic.

4) Linking points 2 and 3 above – I feel that there is a step missing from the computational model, or at least a step missing from the interpretation of the model in the discussion. Confidence is positively related to activity on hits, but negatively related to activity on misses. This implies that there must be some downstream computation that does the inversion – i.e. that “knows” to interpret strong PPC signals as indicating higher confidence when saying “yes”, but lower confidence when saying “no”. Such an architecture would nuance the authors' interpretation of a single code in PPC for both detection and confidence.

Minor points:

- “no-responses” on line 73 is potentially ambiguous as it could mean reporting “no”/“absent” rather than non-report
- Could the base rate of catch trials be reported in Results? I know it’s in methods, but I think it would be useful here for the reader to provide context when interpreting the conservative criterion of the patient.
- What is $v(t)$ in equation 2? It’s not defined in the text and I’m wondering whether it’s even necessary given that $d(t)$ stands for the drift rate
- How was observation noise handled when fitting the model to hit/false alarm rates and EEG data (p. 24)? A Gaussian link function is alluded to but no equations are given. Were these fixed or free parameters? Also I think this passage should say the log-likelihood “corresponded to the log of the normal probability...”

Reviewer #2 (Remarks to the Author):

General

The study by Pereira et al describes single neuron recordings from a microarray in the posterior parietal cortex of a young adult epileptic [redacted] while they perform a simple vibrotactile detection task. In the main experiment, they collect response times (RT) for successful detections (Hits). The RT design delineates the epoch, between the stimulus and the report, in which the decision is made. The downside of RT in this design is that there is no RT for a failures to detect (Misses). In a second experiment the authors obtain explicit “no” responses, but without RT. In this case Hits, Misses, Correct Rejections and False Alarms are discerned along with a 3-level confidence rating. The authors interpret the first experiment as support for an evidence accumulation process with a terminating threshold (or bound) that establishes detection and its timing. They interpret the confidence ratings as support for metacognitive monitoring. The combination of findings are adduced to support a conclusion that threshold applied to the representation of accumulating evidence in parietal neurons explains the transition from unconscious deliberation to a conscious awareness of the vibration (as in Dehaene et al., 2014; Kang et al., 2017).

It is rare to obtain quality single neuron recording from the human brain during well controlled perceptual experiments involving perception, especially accompanied by measures of RT. It is rarer still to do this in a computational framework, and extremely rare to obtain such recordings from the parietal cortex. I am therefore enthusiastic about the dissemination of these data to the scientific community. I will share several reservations about the claims and analyses, but they are intended to help the authors improve what is already an interesting paper. To put it in perspective, it is far superior to a recent publication on this topic in Science. I would ask the authors and the editor to read some of the critical comments below as an attempt to help the authors sharpen the analyses and interpretation.

Major concerns

A. In Experiment 1, the authors propose that their neural recordings are consistent with the accumulation of evidence and noise (aka, deterministic drift plus diffusion) until the neural activity reaches threshold to terminate the decision in a detection report. The evidence for a threshold level of neural activity at termination is reasonably compelling, but the evidence for accumulation is less convincing (e.g., example neurons in Figs 1C-D). The “hallmark of a process leading to a decision threshold (Line 95)” seems promising for the leftmost histogram of the three neurons (Fig 1D), but in panel C, this neuron turns out to simply start its rise later on the trials with longer RT. In drift-diffusion the (1) the start of the accumulation is time locked to the stimulus and (2) the process halts when the accumulation reaches a threshold level just prior to the report. Without #1, observation #2 is consistent with any stereotyped neural response associated with the completion of the decision. Perhaps the problem is only a matter of presentation. The authors state that “the cumulative sum of spikes following stimulus onset correlated with the corresponding RT in 67/94 detection-selective neurons...” (lines 97-98). Perhaps these 67 neurons could be subjected to a more rigorous analysis, especially if they were recorded simultaneously (see below).

B. The authors also exploit a second hallmark of diffusion, which stems from the the accumulation of noise over time. A running sum of independent and identically distributed (iid) samples of noise has a variance that increases linearly with the number of samples and a correlation between cumulants from samples 1 to i and samples 1 to $j > i$ such that $r_{ij} = \sqrt{i/j}$. Conformance to this pattern has been demonstrated in the parietal cortex of monkeys during evidence accumulation. The challenge is to identify the variance of these latent accumulations from the variance of the point process (spike trains), which would contribute variance even if the latent rate were

identical on every trial. These are the statistics that the authors refer to as VarCE and CorCE (the CE stands for 'of the conditional expectation').

It is laudable that the authors attempted to interrogate their data for these signatures, but there are several issues of concern. (1) It is essential to use all trials, not just the hits, and it is crucial to perform the analysis in an epoch that is not affected by many terminations. I don't know if this is possible without knowing the RT for Miss trials. (2) There is a better way to estimate ϕ than the description on line 687 (see Shushruth, J Neurosci 2018). (3) The CorCE must be unity along the main diagonal. That is not what is shown in Fig 1F. (4) There is no point to doing this analysis in neurons that do not show a clear rise in the mean firing rate on Hits. One is seeking a 2nd order confirmation of an accumulation process. If it's not suspected on the basis of a first order observation, what's the point? (5) It is essential to analyze the 1st juxtadiagonal of the CorCE matrix (i.e., lag=1). Decorrelation as a function of lag is nonspecific. It is the combination of this decay with the increase in correlation for lag=1, as a function of time that provides evidence for the process. (6) Unless the authors have many neurons like the three examples, it's unlikely they have the power to discern these 2nd order hallmarks.

It might be worthwhile to give it another try with these concerns in mind. Consider identifying a subset of the 47 neurons; apply a sensible attrition rule; concentrate on data during the accumulation without many terminations (or perhaps guided by Fig 1E, looking only from 300-800 ms); include the Miss trials. Alternatively, why not take advantage of the large number of recorded neurons? VarCE and CorCE provide an indirect glimpse at the latent rates across trials from single neurons, which are hypothesized to represent drift+diffusion, and the diffusion part is suppressed by averaging across trials (the expectation of the diffusion is 0 for all t). A multi-neuron data set might have several neurons that show signs of the drift part of drift+diffusion in the average. If so, one can average across these neurons on a single trial to obtain an estimate of the population rate on that trial. One can then inspect these across trials to observe the diffusion directly.

C. Experiment 2 is a delayed double-response task. Its main contribution to the paper is to bolster the idea that the neural responses are tied in some way to the subjective experience of the detection. We already know that the subject was not guessing ($d' = 1.64$ by my calculation), so we can believe that the subject felt the vibration. The fact that confidence judgments were greater on hits than misses does not say much. It is actually inconsistent with standard signal detection when Hit and Miss rates are equal, assuming (i) a Gaussian distribution for the signal plus noise and (ii) a 'distance from criterion' measure for confidence. My interpretation of the Fig 2A (bottom) is that these two assumptions are incorrect and/or the participant interpreted the scale differently for hits and misses. I think the authors agree. So we learn little from the behavior other than to be skeptical about the metacognitive measures mentioned later in the paper. The crucial issue ought to be the relationship between neural activity and confidence, given the same choice. I'm not sure the GLM (lines 646-650) is useful here. Confidence is heavily weighted to the middle for misses. A significant interaction term might be correcting for this. The authors need to devote more attention to the single neuron analysis here. They should show that on Hits the firing rates covary with confidence rating. It may be that some further subclassification of the neurons is required, and assuming it is justified (eg, neurons that show some modulation for hits in the first palace and do so with the same sign). It weakens the thesis to rely on the decoding analysis, because one must accept that the brain employs such a decoder. It's one thing to state that information is present using a machine-learning statistical procedure. It's another to state that this information is related to subjective monitoring. I raise some specific concerns about Experiments 3 and 4, below, but they are not major. The possible exception is that the use of the Ornstein-Uhlenbeck process in Experiment 4 is a potential source of confusion to readers, as this model seems antithetical to the ideas put forth in Experiments 1 and 2.

D. A final major concern is over the framing of consciousness. I have already stated my position that the data provided by the authors is special and worthy of dissemination. The framing in relation to conscious awareness is valiant. Assuming the authors can rise to the challenges above, they are entitled to interpret their finding as they see fit. That said, I would encourage the authors to steer clear of some common pitfalls. The conversion from preconscious neural activity to a conscious state is likely to involve an operation like a threshold, applied to the representation of information or evidence. But the same is true for many mental operations that do not lead to conscious awareness. For example, "ignition" in Dehaene's global neural workspace theory (GNW) does not explain consciousness, whereas some consequence of the ignition might hold the key. Calling it a GNW is not enough. What we know is that lots of brain is activated, but this is hardly the explanation we are looking for, and by Dehaene's own reckoning, there is no example of an area in the GNW that is not also activated under conditions that support mental operations that are not consciously experienced. The other pitfall is the degree to which one relies on confidence as a sign of "monitoring". The confidence ratings can be understood by multiple criteria (rating scale psychophysics) without any appeal to phenomenology—or at least no more appeal than one would invoke in a simple detection task. I think it is safe to say that correct detections (Hits) were reliably experienced because the human said they were and because their graded confidence is consistent with the process that explains the rate of hits, correct rejects, etc. I don't know how well you can make the latter point. For example the confidence ratings in Experiment 4 do not make sense to me.

Specific concerns

The abstract makes use of 3 terms, perceptual consciousness, reflexive consciousness, and reflexive monitoring. This is confusing. Is reflexive monitoring just another term for perceptual monitoring? Elsewhere, reflexive consciousness also makes an appearance. Please clarify the meanings of these terms.

L21. "...confidence as the distance between maximal evidence and that bound." AND^[SEP]L216. "Confidence was read out from the distance between accumulated evidence and the decision threshold". That does not make sense in evidence accumulation to a stopping bound. It implies that all detections would share the same confidence. See papers by Doug Vickers, Jerome Busemeyer and Roozbeh Kiani.

L47 "...evidence accumulation process reaches a threshold [12] similar to the physiological processes underlying decision-making.[13-16]" Reference 40 is germane to both parts of this sentence. Reference 14 is irrelevant; it presents no support for evidence accumulation, nor is it a reaction time task. Causal evidence for the influence of parietal neurons on choice and RT can be found in Hanks et al, 2006 Nat Neurosci 9: 682 - 89.

Fig 1B. The horizontal dashed line labeled "no" is not described in the figure caption. How does the subject indicate "no"? I was led to believe it was just the absence of a key press by 2 s. The caption to B is confusing overall. More details are needed in Methods about the behavior. For example, what is the operational definition of a false alarm? Would a response at 2.1 s count as a FA or a Miss? Or are there sham trials that are interleaved? If so, does the subject just experience these as longer gaps? Was there feedback given? If so was the or correct and errors? If so, when was the feedback given for a miss and correct reject?

L62 "Yet, to our knowledge the underlying neural mechanisms remain to be described." See Kiani & Shadlen, Science 2009.

L103 The statement about covariance is weak. Many processes exhibit a decrease in correlation as a function of time lag. For diffusion one must also observe an increase in correlation, for fixed lag, as a function of time. See major concern B.

Fig 1E shows the increase in VarCE up to 1 second. That can be explained by a stopping bound.

L187. Experiment 3 is a nice addition. It shows that neurons (some parietal neurons) process the stimulus even when the subject is not engaged in the task. But i don't understand what the term "task activity" means. Also the lack of a report does not mean the brain did not prepare one provisionally.

L212 "A stimulus was perceived if the simulated evidence accumulation (EA) process reached a bound 40 at any time during the stimulus window (Fig. 4A), compatible with all-or-none views of conscious access." This could be said of non-conscious processing too. For example, something flickers in the visual periphery. The parietal cortex may or may not encode a plan to direct attention there. The subject may be unaware but subsequent eye movements might reveal that the object was effectively interrogated. Such gnostic states probably rely on the provisional commitment that makes use of a threshold applied to evidence, but it need not result in consciousness.

L226 Fig 4D, Why is confidence high on the Misses?

The use of the Ornstein-Uhlenbeck process (OUP) in Experiment 4 is confusing. The OUP is not really an evidence accumulation process; it is a leaky accumulation, more like smoothing. Was this used only for the ECoG or do the authors think the electrophysiology in Experiment 1 is also described by such a model? If so the VarCE and CorCE relationships associated with accumulation are misguided. Presumably the author believe the same mathematical process underlies the decisions in both experiments. It so, then the prediction for confidence should not be based on a signal that does not represent the underlying process.

Metacognitive sensitivity (LL 798-802). Please spell out what you calculated and how the numbers relate to the values reported in the text and graphs. It is reasonable to express skepticism about the metacognitive measures (e.g., meta-dprime) because they fail to acknowledge that the distribution of S+N is not the same as the conditional distribution of S+N, given the response. This creates distortions which are further compounded by violations of the Gaussian and variance assumption.

The authors seem to be unaware of older work from David Heeger using fMRI to study a similar detection task (e.g., Ress et al., Nat Neurosci 2000).

Reviewer #3 (Remarks to the Author):

Pereira et al. present an elegant set of complementary experiments, including intracranial single neuron and ECoG recordings from a single patient, and scalp EEG in a group of 18 healthy subjects. They suggest that their experimental and computational model evidence support a neural evidence accumulation account of “perceptual consciousness” and “perceptual monitoring” of a vibrotactile stimulus presented at near-detection level.

Overall, I feel that this paper is technically and conceptually very impressive. The experimental design and data analysis approaches are highly rigorous and take care to appropriately control for important factors (e.g. using delayed and no report paradigm) to support the major findings. The paper is an important contribution to our understanding of mechanisms that give rise to conscious access, importantly providing a computational framework that could be further extended in the future to test generalizability (beyond the multiple experiments already presented here).

My concerns are only minor. I found the narrative to be sometimes difficult to follow, and I think improvements are needed in the flow and the consistency of terminology used. Moreover, the authors largely describe the single neuron, ECoG, and EEG analyses as replications of one another, but these distinct modalities do not capture the same neural phenomena. I hope that the authors can address the following points:

- I think the manuscript title should be changed to “Evidence accumulation determines VIBROTACTILE conscious access” to better acknowledge the scope of the paper

- The ECoG and EEG results are largely described as “replication” or “generalization” of the single neuron findings. Yet in both ECoG and EEG analyses, the authors focus on raw voltage fluctuations, which can reflect electrophysiological processes that are unrelated to neuronal firing. In the domain of ECoG, evidence suggests that high-gamma/high frequency broadband (~50-200 Hz) power amplitude may be more closely related to neuronal population firing near a given electrode contact (see Parvizi & Kastner 2018 Nat Neurosci for recent review). The authors could consider redoing the ECoG analyses based on high-frequency power amplitude. At minimum, the issue should be discussed, and the analyses should not be framed as “replication” (though the similarity of findings across neuron firing, ECoG and EEG is indeed striking as presented).

- In the abstract, the first sentence describes “perceptual consciousness” as encompassing (1) the perceptual experience, and (2) reflexive monitoring. Later, these terms are described as “perceptual and reflexive consciousness” and then throughout the paper, the term “perceptual monitoring” is used (I believe synonymously with reflexive monitoring). The title of the paper refers to “conscious access.” The paper would be much easier to follow if terms are used much more consistently throughout. Relatedly, I recommend referring to the following paper, which distinguishes between access and phenomenal consciousness Block, N. (1995). On a confusion about a function of consciousness. Behavioral and brain sciences, 18(2), 227-247.

- Abstract: “...irrespective of response effectors” is ambiguous here without context (though I understood after reading the whole paper) and I think should be worded more clearly

- The rationales for the three experimental paradigms (immediate response, delayed response, and no report) is not really fully explained until late in the Discussion (second last paragraph, lines 369-372). The use of these multiple paradigms is a major strength of this study, and I think the rationales should be further clarified and explicitly stated in the Intro and better embedded throughout the description of results. In the Intro, a short discussion of how correlative (reported vs. unreported) findings have been suggested to be confounded by postperceptual processes would be helpful to motivate their exp. 2 and exp. 3. I recommend referring to the following paper: Block, N. (2020 Trends Cogn sci). Finessing the bored monkey problem.

- An updated review on global neuronal workspace perhaps could be cited when discussing relevant concepts: Mashour, G. A., Roelfsema, P., Changeux, J.-P., & Dehaene, S. (2020). Conscious processing and the global neuronal workspace hypothesis. Neuron, 105(5), 776-798.

- In the introduction, I recommend that the authors expand on the notion of ignition and the role of evidence accumulation in GWT, as the entire paper is built from these theoretical considerations. To my knowledge, the concept only applies to GWT and the authors should be explicit about this (or argue otherwise). Higher order models of consciousness and recurrent activation theories do not suggest such a mechanism (e.g. Lau, H.

(2019b). Consciousness, Metacognition, & Perceptual Reality Monitoring. PsyArXiv. doi:10.31234/osf.io/ckbyf.

- For the EEG study, it seems that details may be missing regarding how subjects were screened inclusion/exclusion criteria.
- If possible, the authors should report on the intracranial patient's state at the times of testing (e.g. medications, pain, cognitive state/mood), especially since results are based on a single case.
- Please add information about the size of ECoG electrode contacts
- In Fig 1D, the "Slow/Mid/Fast" text is partly cut off

REVIEWER COMMENTS

Reviewer #1 (Remarks to the Author):

This paper reports analyses of a very rare and valuable dataset of single neuron recordings from human posterior parietal cortex during a near-threshold detection task. The authors were able to relate the same accumulation-to-bound signals observed in single neurons to EEG recordings in healthy volunteers, and show that a computational model of evidence accumulation is able to accommodate both the neural responses and the confidence data. Perhaps most impressively, this model was able to predict confidence levels in the absence of a detection response (i.e. on miss trials) in the patient data, suggesting subtle modulations of confidence in saying “no stimulus was present” depend on the status of subthreshold evidence accumulation in the parietal cortex.

This is a tour de force combining different sets of neuronal data with an elegant and simple computational model to account for variation in perceptual confidence. I have three main comments that I hope the authors will find useful, together with minor questions about the modelling approach. I have limited knowledge of single unit analysis so my review mostly focuses on the psychophysics and interpretation of the model.

We thank the reviewer for the kind words and constructive comments. We hope that the reviewer will appreciate this revised version of the manuscript and the new analysis revealing neurons encoding misses.

1) I found the introduction somewhat skewed by interpreting detection through the lens of perceptual consciousness. I am on board with the idea that studying detection+confidence can give us a particularly sensitive window on the computations supporting visual awareness. But I found the current framing to fall short in making this case. In particular, there is a notable failure to cite classic single-unit studies of detection in macaques – for instance Cook & Maunsell (2002) Nat Neurosci for detection of coherent motion, de Lafuente & Romo (2005) Nat Neurosci for detection of vibrotactile stimuli and Merten & Nieder (2012) PNAS for detection of near-threshold contrast. Many of these studies I think align with the authors’ findings (of a graded accumulation that varies with stimulus intensity) albeit in different brain regions. However, and perhaps this is why they weren’t included here, they are notably silent on the issue of consciousness, and instead simply present their results as neural correlates of psychophysical detection curves. I think the authors do have a strong case for framing their study as novel in this context – because the recordings are in humans, and subjective confidence reports are elicited, they offer an unusual window onto whether similar evidence accumulation processes that have been discovered in monkeys might also support perceptual awareness in humans. But the way the paper is currently

structured gives the reader the impression that such studies haven't yet been done at all, in animals or humans (cf. lines 48-50).

Although we were careful to refer to animal work with respect to evidence accumulation in discrimination tasks, it is true that we missed some important references concerning neuronal activity related to detection. We acknowledge and regret that the introduction could reflect an unfair sense of novelty with respect to animal works and have revised it accordingly, including the references suggested by the reviewer.

We added the following sentence (lines 44-45).

"[...] and several animal studies have reported single neuron correlates in detection tasks (Cook & Maunsell, 2002, Nat. Neuro; de Lafuente & Romo, 2005, Nat. Neuro.; Merten & Nieder, 2012, PNAS)."

We hope that this addition gives a fairer representation of the field, combining animal studies using detection tasks, and human studies using detection, confidence, and no-report paradigms.

2) I found it striking that across all experiments (and also the EEG data), misses were effectively default-coded in terms of evidence accumulation. In other words, stimuli that were missed or otherwise of low intensity led to the lowest average firing rates. This makes sense in relation to a graded magnitude code for intensity. But it is notably at odds with the notion that neural populations might actively code for each response category – i.e. an active coding of both “yes” and “no” responses (see Merten & Nieder, 2012 PNAS for findings on the active coding of stimulus absence in macaque PFC independent of report). Did the authors discover any neural responses in PPC that actively coded for “no” responses / confident misses? Would these have been apparent in the current analysis if they had been present? If there are none, I wonder whether this is actually an informative null result – suggesting that parietal neurons provide a low-dimensional magnitude code that might then be “read out” by the kind of PFC populations identified by Merten & Nieder, to actively code for presence and absence.

This is a very good point, and we thank the reviewer for raising it. In the original manuscript, we only reported the contrast between hits and misses; neurons with a higher firing rate for misses would thus be included in the analysis but we did not specifically look for them. Following the reviewer's implicit suggestion, we dug deeper in the matter and were pleasantly surprised to find what is described hereafter:

We searched for miss-coding neurons, concentrating on experiment 1 which had more detection-selective neurons to be stratified into different categories. To do so, we relied on a two-step categorization approach. As a first step, we identified detection-selective neurons which had significantly different spike rates between the baseline (0.3 s pre-stimulus) and the window of interest (0.5 - 1.5 s post-stimulus). We found 24 such neurons (permutation test: $p = 0.001$), with 16 showing a significant increase in spike rate compared to baseline ($p = 0.001$). As the reviewer may notice, these neurons may not necessarily encode misses specifically, since they could have a higher firing rate for misses compared to baseline but an even higher firing rate for hits. In a second step, we thus further subcategorized these neurons into *miss-coding* neurons with a higher firing rate for misses compared to both baseline and hits ($N = 6$ neurons, $p = 0.001$). We interpret such firing rate patterns as ‘actively coding for “no” responses’. Of note, although we found only six of these neurons, which calls for cautious interpretations, the criteria to define them were extremely conservative. We found no miss-coding neurons in Exp. 2, possibly because there were five times less detection-selective neurons (17 instead of 81).

We reasoned that this analysis also called for some additional analysis for possible increases in firing rates for hits. We now report these additional analyses in the result section (lines 94-96 and 103-108):

44 / 81 of these detection-selective neurons significantly increased their firing rate compared to a 300 ms baseline prior to stimulus onset (Supplementary Fig. 2b; $p = 0.001$) while only one decreased its firing rate ($p = 0.78$). [...]

Regarding misses, 16 / 81 neurons significantly increased their firing rate compared to baseline (Supplementary Fig. 2c; $p = 0.001$). Average firing rates for these 16 neurons suggested that this increase from baseline, was lower than the one we observed for hits. We therefore sought for neurons that increased their firing rates for misses and had higher spike counts for misses than for hits. We found 6 such miss-coding neurons (Fig. 1g, Supplementary Fig. 2d; $p = 0.001$), suggesting that some neurons in the PPC actively code missed stimuli (Merten & Nieder, 2012, PNAS).

We also added a Supplementary Figure to illustrate the spike rate pattern of these neurons:

Supplementary Fig 2. Average firing rates of different types of responsive neurons in Experiment 1. Firing rates were normalized using a 0.3 s pre-stimulus baseline, for three bins of RT (hits; blue) and for misses (red). (a) Detection-selective neurons, (b) Subset of a with an increase in firing rate compared to baseline. (c) Subset of a with an increase in firing rate for misses but not as high as for hits. The increase in firing rate for misses can be seen for the red trace. (d) Subset of a with an increase in firing rate for misses which was stronger than for hits. All shaded areas represent 95%-CI.

Finally, we discuss possible roles of miss-coding neurons (lines 360-364):

We found six neurons showing a higher increase in firing rate for misses compared to hits, raising the intriguing possibility that some neurons could respond specifically to misses. This extends a study on non-human primates which found prefrontal neurons also coding for misses but rather in the delay period following the stimulus (Merten & Nieder, 2012, PNAS). However, more work is needed to show how such neurons could contribute to confidence in detection tasks.

It is worth noting that in the Merten & Nieder paper, “no” response neurons were only found in a delay period starting 1.9 s after stimulus onset. We did find neurons that had different firing rates for hits and misses during the delay period of experiment 2. However, if the reviewer agrees, we will not include this analysis in the manuscript, as these neurons represent a purely post-perceptual process while our paper focuses on perceptual consciousness.

3) It took me a while to figure out how the model in Figure 4 accounted for confidence in misses. As far as I understand it from the methods, miss-confidence is inverted, so that the lower the firing rate peak, the more confident I am that nothing is there. That makes sense – but it’s difficult to infer from Figure 4A. Can a similar graphical interpretation be added for miss-confidence, perhaps as a separate panel? How is this defined in relation to threshold? In relation to this point, I was initially surprised to see that the correlation coefficient between miss-confidence and activity was negative (line 263), but eventually figured out this makes sense under a definition in which miss confidence is inversely related to activity. It would be useful to help the reader through this logic.

The reviewer is correct in assuming that confidence is inverted with respect to the decision threshold for “no” responses (i.e. misses and correct rejections). We agree that this was not clear enough in our original submission, and we added the proposed panel (Fig. 4b) to dissipate any doubts regarding our model assumptions.

Fig. 4. Computational model based on evidence accumulation. (a) Simulation: Time-varying drift rate (d ; thick black trace) had a short-lasting boost after a non-decision time following stimulus onset (dashed vertical line). Example evidence accumulation for one trial (EA; cyan trace for a hit, red trace for a miss) rises sharply after the drift boost and is attracted back to zero due to leakage. A stimulus is considered as perceived (hit) if EA reaches a decision threshold (horizontal line), and as non-perceived (miss) if not. (b) The maximum of accumulated evidence with respect to the decision threshold is used as a confidence readout. In the example in (a), the cyan trace results in high confidence in a hit (large distance from the bound) and the red trace results in low confidence in a miss (small distance from the bound).

As for the correlations, we hesitated while writing the manuscript on whether we should invert confidence to have positive correlations but choose to stay as close as possible to the data. We therefore updated the text to make this clear:

Lines 227-229:

Confidence was read out from the difference between accumulated evidence and the decision threshold, with a sign inversion for misses and correct rejections, since the closer the evidence to the bound, the lower the confidence.

4) Linking points 2 and 3 above – I feel that there is a step missing from the computational model, or at least a step missing from the interpretation of the model in the discussion. Confidence is positively related to activity on hits, but negatively related on misses. This implies that there must be some downstream computation that does the inversion – i.e. that “knows” to interpret strong PPC signals as indicating higher confidence when saying “yes”, but lower confidence when saying

“no”. Such an architecture would nuance the authors’ interpretation of a single code in PPC for both detection and confidence.

Indeed, although our model implies an inversion to compute confidence in misses, our neural data provides no direct evidence for such a process. One possibility is indeed that this inversion takes place at a later stage in prefrontal areas. We now briefly mention this issue as a venue for future research in the revised discussion.

Lines 350-352: It remains unknown however, where this readout is implemented, as we had limited anatomical coverage in this individual patient, and did not find parietal neurons encoding confidence irrespective of detection.

Minor points:

- “no-responses” on line 73 is potentially ambiguous as it could mean reporting “no”/“absent” rather than non-report

We agree with the reviewer, and changed to “no-report” accordingly.

- Could the base rate of catch trials be reported in Results? I know it’s in methods, but I think it would be useful here for the reader to provide context when interpreting the conservative criterion of the patient.

Unless we misunderstood the reviewer’s point, we reported this information in the results section, on lines 131-132 of the original manuscript: “20% trials had no stimuli”. We added “(catch trials)” for clarity.

- What is $v(t)$ in equation 2? It’s not defined in the text and I’m wondering whether it’s even necessary given that $d(t)$ stands for the drift rate

We are sorry for this, we replaced $v(t)$ by its expression (line 790).

- How was observation noise handled when fitting the model to hit/false alarm rates and EEG data (p. 24)? A Gaussian link function is alluded to but no equations are given. Were these fixed or free parameters?

We used a Gaussian link function with fixed standard deviation. For hit and false alarm rate we used a standard deviation of 0.02, corresponding to a confidence interval of around $\pm 4\%$, which worked better than a logit. To fit EEG data, we used a standard deviation of 0.2 which corresponded to a confidence interval of around $\pm 0.4 \mu\text{V}$. We found that these values corresponded to a reasonable range considering the corresponding behavior and EEG. In a previous attempt to fit the model, we used the variability of the EEG directly, but the fit was worse. However, our results on confidence still held. Therefore, we do not think that these parameters affect our conclusions. We have adapted the methods section accordingly (lines 813-815).

The standard deviation of the observation noise was set to 0.02 for hit-rate and false-alarm rate and 0.2 for $EA(t)$ and $pEA(t)$.

Also I think this passage should say the log-likelihood “corresponded to the log of the normal probability...”

The reviewer is right, we corrected this accordingly.

Reviewer #2 (Remarks to the Author):

General

The study by Pereira et al describes single neuron recordings from a microarray in the posterior parietal cortex of a young adult epileptic [redacted] while *they* perform a simple vibrotactile detection task. In the main experiment, they collect response times (RT) for successful detections (Hits). The RT design delineates the epoch, between the stimulus and the report, in which the decision is made. The downside of RT in this design is that there is no RT for a failures to detect (Misses). In a second experiment the authors obtain explicit “no” responses, but without RT. In this case Hits, Misses, Correct Rejections and False Alarms are discerned along with a 3-level confidence rating. The authors interpret the first experiment as support for an evidence accumulation process with a terminating threshold (or bound) that establishes detection and its timing. They interpret the confidence ratings as support for metacognitive monitoring. The combination of findings are adduced to support a conclusion that threshold applied to the representation of accumulating evidence in parietal neurons explains the transition from unconscious deliberation to a conscious awareness of the vibration (as in Dehaene et al., 2014; Kang et al., 2017).

It is rare to obtain quality single neuron recording from the human brain during well controlled perceptual experiments involving perception, especially accompanied by measures of RT. It is rarer still to do this in a computational framework, and extremely rare to obtain such recordings from the parietal cortex. I am therefore enthusiastic about the dissemination of these data to the scientific community. I will share several reservations about the claims and analyses, but they are intended to help the authors improve what is already an interesting paper. To put it in perspective, it is far superior to a recent publication on this topic in Science. I would ask the authors and the editor to read some of the critical comments below as an attempt to help the authors sharpen the analyses and interpretation.

We thank the reviewer for their enthusiasm despite their legitimate reservations, which we have addressed to the best of our capacity. We hope that they will find this revised version with adapted claims/interpretations, new analyses and acknowledged limitations convincing, keeping in mind that some results that are normative for discrimination tasks based on random dot kinematograms might not hold for detection tasks based on vibrotactile stimulation with unpredictable onsets.

Major concerns

A. In Experiment 1, the authors propose that their neural recordings are consistent with the accumulation of evidence and noise (aka, deterministic drift plus diffusion) until the neural activity reaches threshold to terminate the decision in a detection report. The evidence for a threshold level of neural activity at termination is reasonably compelling, but the evidence for accumulation is less convincing (e.g., example neurons in Figs 1C-D). The “hallmark of a process leading to a decision threshold (Line 95)” seems promising for the leftmost histogram of the three neurons (Fig 1D), but in panel C, this neuron turns out to simply start its rise later on the trials with longer RT. In drift-diffusion the (1) the start of the accumulation is time locked to the stimulus and (2) the process halts when the accumulation reaches a threshold level just prior to the report. Without #1, observation #2 is consistent with any stereotyped neural response associated with the completion of the decision. Perhaps the problem is only a matter of presentation. The authors state that “the cumulative sum of spikes following stimulus onset correlated with the corresponding RT in 67/94 detection-selective neurons...” (lines 97-98). Perhaps these 67 neurons could be subjected to a more rigorous analysis, especially if they were recorded simultaneously (see below).

We agree with the reviewer’s concern. It is true that our method of correlating RT with the cumulative sum of spikes was suboptimal in selecting neurons with gradual increases in firing rate, since it did not concentrate specifically on the time preceding the response. We followed the reviewer’s suggestion and applied a more rigorous stimulus-locked analysis:

For each neuron and trial, we fit a regression line to the firing rate between 300 ms post-stimulus and the response onset. We then select neurons for which the so-computed slope significantly correlates with the RT ($p < 0.05$; Spearman correlation). This stimulus-locked analysis thus specifically targets the pre-decisional period and was much more conservative (resulting in the selection of 14 neurons instead of 67, which remains significantly higher than what we could expect by chance: $p = 0.001$). It selected the leftmost neuron but not the two right ones, confirming the reviewer’s intuition. Overall, these results confirm that the start of the accumulation is time locked to the stimulus. We updated the methods (lines 698-670), the main text (lines 96-102) and Figure 1 panels d and e and added a panel with firing rates averaged across selective neurons, time-locked to the stimulus onset or to the response onset (see below).

Figure 1. (f) Average firing rates for detection- and RT- responsive neurons, for three bins of RT (hits; blue) and for misses (red). Firing rates were normalized to a 0.3 s pre-stimulus baseline and time-locked to the stimulus onset. Inset: Corresponding average firing rates time-locked to the response

In our view, the qualitative assessment of evidence accumulation for single neurons is hindered by the fact that most neurons had lower firing rates than typical LIP neurons in non-human primate studies, likely because we had no way of tuning the electrode position after surgery for clinical purposes. We also note that in the non-human primate literature, the stereotypical firing rates for evidence accumulation are often averaged across all recorded neurons (e.g. Roitman & Shadlen, 2002, *J. Neuro*; Shushruth et al., 2018, *J. Neuro*).

To provide further evidence (and further reassure ourselves and the reviewer), we performed the same analysis this time combining single and multi-units. We thereby included 11 clusters that were identified by the spike sorting algorithm but not considered as coming from a single unit (e.g. because the percentage of inter spike interval was too high). These results seem to be closer to what is expected from an evidence accumulation process (Supplementary Fig. 3a). Moreover, the analysis the reviewer suggested in the next concern provides further evidence for the fact that “(1) the start of the accumulation is time locked to the stimulus.”.

We also provide the same panel with five bins of RT to show that this pattern is consistent across several parametrizations (Supplementary Fig. 3b,c). We notice that for the first two bins of RT, evidence accumulation seems to start earlier. This is consistent with Fig. 5a in the study from Cook & Maunsell, 2002, (*Nat. Neuro*) which also used a detection task. As the reviewer can see in our new Supplementary Fig 8 (see below), we were also able to reproduce this qualitatively with a computational model of evidence accumulation.

Of note, we added traces of the firing rates after the RT. As we argue below in response to the reviewer's point below, we consider that these are consistent with post-decisional evidence accumulation.

We included these results in the supplementary materials.

Supplementary Fig 3. Average firing rates of detection- and RT-selective single- and multi-units, combined (Experiment 1). Firing rates are normalized using a 0.3 s pre-stimulus baseline, for three bins of RT (hits; blue) and for misses (red). (a) Aligned to stimulus onset for three bins of RT (hits; blue) and for misses (red). (b) Aligned to stimulus onset for five bins of RT (hits; blue) and for misses (red). Traces without shaded areas represent firing rates after the median RT for each bin and suggest that evidence accumulation continues after the decision. (c) Firing rates for the four RT bins of b, aligned to the response time (RT).

B. The authors also exploit a second hallmark of diffusion, which stems from the the accumulation of noise over time. A running sum of independent and identically distributed (iid) samples of noise has a variance that increases linearly with the number of samples and a correlation between cumulants from samples 1 to i and samples 1 to $j > i$ such that $r_{\{ij\}} = \sqrt{i/j}$. Conformance to this pattern has been demonstrated in the parietal cortex of monkeys during evidence accumulation. The challenge is to identify the variance of these latent accumulations from the variance of the point process (spike trains), which would contribute variance even if the latent rate were identical on every trial. These are the statistics that the authors refer to as VarCE and CorCE (the CE stands for 'of the conditional expectation').

It is laudable that the authors attempted to interrogate their data for these signatures, but there are several issues of concern. (1) It is essential to use all trials, not just the hits, and it is crucial perform the analysis in an epoch that is not affected by many terminations. I don't know if this is possible without knowing the RT for Miss trials. (2) There is a better way to estimate ϕ than the description on line 687 (see Shushruth, J Neurosci 2018). (3) The CorCE must be unity along the main diagonal. That is not what is shown in Fig 1F. (4) There is no point to doing this analysis in neurons that do not show a clear rise in the mean firing rate on Hits. One is seeking a 2nd order confirmation of an accumulation process. If it's not suspected on the basis of a first order observation, what's the point? (5) It is essential to analyze the 1st juxtadiagonal of the CorCE matrix (i.e., lag=1). Decorrelation as a function of lag is nonspecific. It is the combination

of this decay with the increase in correlation for lag=1, as function of time that provides evidence for the process. (6) Unless the authors have many neurons like the three examples, it's unlikely they have the power to discern these 2nd order hallmarks. It might be worthwhile to give it another try with these concerns in mind.

Consider identifying a subset of the 47 neurons; apply a sensible attrition rule;

concentrate on data during the accumulation without many terminations (or perhaps guided by Fig 1E, looking only from 300-800 ms); include the Miss trials.

Alternatively, why not take advantage of the large number of recorded neurons? VarCE and CorCE provide an indirect glimpse at the latent rates across trials from single neurons, which are hypothesized to represent drift+diffusion, and the diffusion part is suppressed by averaging across trials (the expectation of the diffusion is 0 for all t).

A multi-neuron data set might have several neurons that show signs of the drift part of drift+diffusion in the average. If so, one can average across these neurons on a single trial to obtain an estimate of the population rate on that trial.

One can then inspect these across trials to observe the diffusion directly.

We highly appreciate this thorough feedback of our variance analysis. We realize that we might have overlooked some important assumptions and, considering that our earliest RTs are around 300 ms, it will be difficult to find an analysis window that does not contain many terminations without removing too many trials. We are therefore grateful to the reviewer for proposing an alternative and more tractable analysis.

We analyzed the aforementioned set of 25 single- and multi-units following the reviewer's suggested analysis plan. We averaged firing rates across single- and multi-units for each trial and displayed the resulting average by increasing RT. We were able to confirm that firing rates start increasing around 300 ms after stimulus onset (solid vertical white line), which happens to correspond to the lowest RTs. We also confirmed that the slope of the average firing rate between 300 ms and the response correlated with the RT, although admittedly, this analysis is circular since we select the units based on this criterion.

We have added this analysis as Supplementary Fig. 3, which confirms that selected units using more stringent criteria do represent an evidence accumulation process. These results also support the fact that the neurons we record start accumulating evidence at a latency time-locked to the stimulus onset.

Supplementary Fig 3. (d,e) Firing rates for single trials, averaged across the 25 detection- and RT-selective single- and multi-units, reordered by RT for the two sessions of Experiment 1 (d and e). Insets represent the slope of the firing rates between 300 ms and the RT, as a function of the RT. All shaded areas represent 95%-CI. Of note, this is the only figure in this study including multi-unit data.

C. Experiment 2 is a delayed double-response task. Its main contribution to the paper is to bolster the idea that the neural responses are tied in some way to the subjective experience of the detection. We already know that the subject was not guessing ($d' = 1.64$ by my calculation), so we can believe that the subject felt the vibration.

The fact that confidence judgments were greater on hits than misses does not say much. It is actually inconsistent with standard signal detection when Hit and Miss rates are equal, assuming (i) a Gaussian distribution for the signal plus noise and (ii) a 'distance from criterion' measure for confidence.

We agree that SDT predicts identical confidence levels when hit-rate is 50% under the two assumptions mentioned by the reviewer, and a third assumption, which is that the confidence criteria used to bin confidence into three levels are symmetrical around the decision criterion. This last assumption is not common in the literature and confidence criteria are usually free parameters in current fitting toolboxes (e.g. Maniscalco & Lau, 2012, *Consc. & Cog.*; Fleming, 2017, *Neurosci Consc*). Note that we did not use these methods in the current manuscript, we refer to them to build our argument.

We nonetheless removed the phrase "[...] indicative of accurate detection monitoring processes", line 143.

My interpretation of the Fig 2A (bottom) is that these two assumptions are incorrect and/or the participant interpreted the scale differently for hits and misses. I think the authors agree. So we learn little from the behavior other than to be skeptical about the metacognitive measures mentioned later in the paper.

We agree that the participant may have interpreted the scale differently for hits and misses, which could translate into asymmetrical confidence criteria around the decision criterion. With this property in mind, a simple simulation shows that the two assumptions mentioned by the reviewer could still be true. Indeed, using a normal distribution representing the difference between the internal perceptual representation X and the decision criterion c , it was easy to find confidence criteria that approximately but reliably reproduced our data without dropping either assumption (and using an identical number of trials).

A. The normal distribution and the (manually) selected confidence criteria.

B. The detection rate was reproduced by our simulations (black dots)

C. Similar distribution shapes were reproduced by our simulations (black traces) for confidence, leading to a higher confidence for hits compared to misses.

That said, we thought and still think that behavioral results from this rare patient deserved to be reported for the sake of completeness/transparency. We understand the reviewer's reservation for SDT-based measures of metacognition. However, as we detail later, our measure of metacognition based on areas under the receiver operating characteristic (AUROC) curve is non-parametric (although it can also be analyzed in an SDT framework).

The crucial issue ought to be the relationship between neural activity and confidence, given the same choice. I'm not sure the GLM (lines 646-650) is useful here. Confidence is heavily weighted to the middle for misses. A significant interaction term might be correcting for this. The authors need to devote more attention to the single neuron analysis here. They should show that on Hits the firing rates covary with confidence rating.

It may be that some further subclassification of the neurons is required, and assuming it is justified (eg, neurons that show some modulation for hits in the first palace and do so with the same sign).

We thank the reviewer for raising this point. However, we only partially agree. We would like to remind the reviewer that since we used nonparametric permutation tests, the class imbalance for the misses could not have induced a statistical bias in the selection of the neurons, since this bias would also have been present in the permuted data. Nonetheless, we agree that the class imbalance calls for caution. When performing the analysis proposed by the reviewer, we found 19/86 neurons with a firing rate correlating with confidence for hit trials (permutation test: $p = 0.001$). However, we still think that the GLM approach is valuable to disentangle the detection and confidence factors, so we conducted further simulation tests.

We simulated spike counts using a Poisson distribution, with a baseline rate of 10 Hz under four situations: i) no effect of detection nor effect of confidence (0), ii) an effect of detection (D), iii) an effect of confidence (C), iv) an interaction between detection and confidence (D x C). Situations ii-iv) were represented by effect sizes of 0.2. We fitted a GLM on these simulated data and compared the results with the approach consisting in testing for differences between hits and misses, and assessing the correlation with confidence in hits. We repeated this process 200 times using the same trial structure as in our data (thus reproducing the class imbalance in confidence for misses).

The first row shows the theoretical spike rate under different situations (columns: 0: no effect, D: effect of detection, C: effect of confidence, D x C: interaction effect). Whiskers represent standard deviation. Hits and misses are slightly shifted for display purposes. The second row shows the proportion of times that our simulated data led to a significant GLM coefficient. Whiskers represent 95%-confidence intervals (CI). The third row shows the proportion of times that the single tests led to significant effects for detection (D), confidence in hits (C(hit)) and both detection and confidence in hits (D & C(hit)). Whiskers represent 95%-CI.

First, these simulations showed that the GLM does not seem to compensate for confidence imbalance for misses, as we found approximately 5% of false positives for the no-effect situation (leftmost column, second line). Importantly, for interaction effects (rightmost column), the alternative approach found effects of detection and confidence, but underestimated the overlap (i.e., hits > misses and correlation with confidence in hits). This is understandable, since obtaining the intersection of detection and confidence in hits underlies multiplying their probabilities. Of note, we found similar results in our neuronal data: when applying successive tests, we found detection and confidence in hits neurons but few neurons that showed both effects. In view of this analysis, we believe that it does not represent evidence against an overlap of detection and confidence in hits effects.

On the contrary, the GLM did a decent job in finding interaction effects. One downside of the GLM was that it was less sensitive in finding detection effects compared to the single test, which would lead us to rather underestimate than overestimate. In sum, this simulation analysis supports the GLM as a better approach to characterize the underlying firing rate of the neurons we recorded.

We obtained similar simulation results when changing baseline firing rates, effect sizes, or when coding the variables so as to have a decrease in firing rate for misses in the interaction effect situation.

One important point that we realized thanks to the reviewer's comment is that our interpretation of the interaction term as an overlap between detection and confidence was a bit far-fetched and we agree that a further subclassification of neurons is needed. To avoid the statistical issues mentioned above, we performed a supplementary analysis on choice probabilities and rank correlation between spike counts and confidence for hits or for misses. We now show a linear relation between choice probability for detection and the correlation between confidence in hits and spike rate, at the population level ($R = 0.54$, $p = 0.001$). This analysis does not require a statistical threshold and is thus, in our view, more adapted to support our claim. We have added the choice probability analysis in the SI (Supplementary text and Supplementary Fig. 4).

Supplementary Fig 4. Choice probability and correlation analysis. Each panel depicts individual neurons, as a function of different measures: (a) Correlation between confidence in hits and spike rate (x-axis) and choice probability (y-axis). (b) Correlation between confidence in hits and spike rate (x-axis) and choice probability (y-axis). (c) Correlation between confidence in hits and spike rate (x-axis) and correlation between confidence in misses and spike rate (y-axis). Regression lines are represented in red with shaded areas representing 95%-CI. Unresponsive neurons are in dark green and responsive neurons in light green (neurons with an interaction effect between detection and confidence). Neurons with a number of spikes correlating with confidence in hits (respectively misses) are circled in cyan (resp. red).

We updated the results section (lines 147-155):

However, 17/86 neurons showed an interaction between detection and confidence (19.77%, $p = 0.02$, permutation test) driven by firing rates for hits with high confidence (Fig. 2b). Among these 17 neurons, we found 9 neurons showing a significant correlation between spike counts and confidence in hits ($p = 0.001$). The sign of this correlation was consistent with the choice probability in 8 of these 9 neurons, suggesting that evidence for detection increases confidence in hits. We found no neuron showing a significant correlation between spike count and confidence

in misses. The same pattern was observed in the population of 86 neurons, as higher correlation coefficients between spike counts and confidence were positively related with choice probabilities for hits but not for misses (Supplementary Fig. 4a–b).

It weakens the thesis to rely on the decoding analysis, because one must accept that the brain employs such a decoder. It's one thing to state that information is present using a machine-learning statistical procedure. It's another to state that this information is related to subjective monitoring.

This is a fair point, which applies to the vast majority of neuroscientific studies employing machine learning. To our defense, the decoders we relied on are just weighted averages of neuronal firing rates. We hope that the reviewer agrees with us that such decoders represent a rather straightforward neural implementation. Although the weights will certainly differ, it does not seem implausible to us that some neurons aggregate the activity from other evidence accumulating neurons. Nonetheless, we have mentioned this issue in the discussion, acknowledging that this *“result implies the existence of neurons aggregating the output of evidence accumulating neurons, which remain to be described”* (lines 324-325). We also added a schema of these decoders in Fig. 2c.

I raise some specific concerns about Experiments 3 and 4, below, but they are not major. The possible exception is that the use of the Ornstein-Uhlenbeck process in Experiment 4 is a potential source of confusion to readers, as this model seems antithetical to the ideas put forth in Experiments 1 and 2.

D. A final major concern is over the framing of consciousness. I have already stated my position that the data provided by the authors is special and worthy of dissemination. The framing in relation to conscious awareness is valiant. Assuming the authors can rise to the challenges above, they are entitled to interpret their finding as they see fit. That said, I would encourage the authors to steer clear of some common pitfalls. The conversion from preconscious neural activity to a conscious state is likely to involve an operation like a threshold, applied to the representation of information or evidence. But the same is true for many mental operations that do not lead to conscious awareness. For example, “ignition” in Dehaene’s global neural workspace theory (GNW) does not explain consciousness, whereas some consequence of the ignition might hold the key. Calling it a GNW is not enough. What we know is that lots of brain is activated, but this is hardly the explanation we are looking for, and by Dehaene’s own reckoning, there is no example of an area in the GNW that is not also activated under conditions that support mental operations that are not consciously experienced. The other pitfall is the degree to which one relies on confidence as a sign of “monitoring”. The confidence ratings can be understood by multiple criteria

(rating scale psychophysics) without any appeal to phenomenology—or at least no more appeal than one would invoke in a simple detection task. I think it is safe to say that correct detections (Hits) were reliably experienced because the human said they were and because their graded confidence is consistent with the process that explains the rate of hits, correct rejects, etc. I don't know how well you can make the latter point. For example the confidence ratings in Experiment 4 do not make sense to me.

We did our best to steer clear from several confounding factors that contaminate studies on consciousness, and to rise to the challenges mentioned above. Yet, and if both the editor and reviewer agree, we would like to avoid taking position regarding the specificity of GNW in the manuscript, as it clearly is beyond the scope of our work. We are aware that different views co-exist regarding broadcasting, some considering it as a neural correlate of consciousness while others as a marker of high-level cognition. Yet, the mechanism we describe in the present study is agnostic with respect to which specific theoretical framework may explain consciousness. We may have over-emphasized GNW in the current manuscript because its link with evidence accumulation is clear, we made an effort to correct this in the revised version, (lines 410-413):

Furthermore, conceptual work will be needed to assess to what extent evidence accumulation is necessary if not sufficient for consciousness (Vlassova et al, 2014, PNAS; Barbosa et al., 2017, Consc. & Cog.), and how it fits within other theoretical frameworks such as higher order thought theories (Brown et al., 2019, TiCS) or recurrent processing theories of consciousness (Lamme, 2006, TiCS).

Importantly, we do not consider that evidence accumulation is both necessary and sufficient for consciousness, and we acknowledge that the thresholding of accumulated evidence is also at play in other domains. In fact, there is even behavioral evidence suggesting that EA can operate unconsciously (Vlassova et al., 2014, PNAS; Barbosa et al., 2017, Consc. & Cog.). We clarified this in the revised discussion by stating that *“stimulus detectability involves the accumulation of sampled evidence towards a decision bound, as previously discussed for discrimination tasks as well as several cognitive functions“*.

Now regarding confidence: we are aware of the theoretical difficulties to use confidence as a reliable test for consciousness (e.g., Rosenthal, 2019, *Neuropsychologia*). Our goal was to propose a common mechanism for both detection reports and confidence, then argue for links between detection and subjective experience. We did not intend to equate confidence ratings with phenomenology, although we acknowledge that our use of different terms related to consciousness was confusing.

As of today, confidence is considered as a gold-standard in human studies of metacognition/self-monitoring (Rahnev et al., 2020, *Nat. Hum. Beh.*), which in turn, can be related to some aspects of perceptual consciousness (Dehaene et al., 2017, *Science*; Brown et al., 2019, *TICS*; Shea & Frith, 2019, *TICS*). We thus used confidence ratings conditioned to first-order detection reports as a proxy to investigate second-order representations and metacognitive monitoring, while avoiding to directly equate those to phenomenology. For example, we wrote in the conclusion: *"We argue that this neuronal mechanism involving a decision bound may serve as a trigger for the neural ignition underlying consciousness and explains how contents remaining inaccessible to consciousness may still be subject to self-monitoring"*. Of note, the derivation of confidence as the maximal evidence accumulated over time suggests the existence of post-decisional mechanisms, which supports the view that confidence reflects monitoring rather than first-order processes.

Finally, regarding confidence in Experiment 4: we were also puzzled at first. However, in preliminary analyses, we found that this difference could be reproduced by an SDT model in which confidence was the weighted distance between the evidence and the criterion after applying a logit function (note that the hit-rate was slightly lower than 50% (SI: 37.81% \pm 1.70 of all trials for hits vs. 42.07% \pm 1.67 of all trials for misses; rate of catch trials: 20%).

Specific concerns

The abstract makes use of 3 terms, perceptual consciousness, reflexive consciousness, and reflexive monitoring. This is confusing. Is reflexive monitoring just another term for perceptual monitoring? Elsewhere, reflexive consciousness also makes an appearance. Please clarify the meanings of these terms.

The reviewer is correct that we considered reflexive monitoring and perceptual monitoring as the same process, which is confusing. We now use only "perceptual consciousness" when referring to subjective experience, and "perceptual monitoring" for self- or metacognitive monitoring. Both these terms are defined in the introduction. We removed any other mention of the term "reflexive".

L21. "...confidence as the distance between maximal evidence and that bound." AND L216. "Confidence was read out from the distance between accumulated evidence and the decision threshold". That does not make sense in evidence accumulation to a stopping bound. It implies

that all detections would share the same confidence. See papers by Doug Vickers, Jerome Busemeyer and Roozbeh Kiani.

We understand that opting-out of a discrimination task is well explained by computing the log-odds of two accumulators representing each choice (Kiani & Shadlen, 2009, Science). If evidence accumulation stops immediately after the decision, this is probably the best way to compute confidence. However, we interpret our neuronal data in a way that departs a bit from the current literature on LIP data.

Indeed, there is strong computational (e.g. Resulaj et al., 2009, Nature) and electrophysiological (e.g. Murphy et al., 2015, eLife) evidence for post-decisional evidence accumulation. Various studies in humans (Fleming et al., 2018, Nat. Neuro.; Van den Berg et al, 2016, eLife; Pereira et al., 2020, PNAS) rely on a post-decisional interpretation of confidence ratings and many works have modeled confidence using a post-decisional readout, including some authors referred to by the reviewer (Pleskac & Busemeyer, 2010, Psych. Rev.). We now also show average firing rates time-locked to the response onset in Fig. 1f and Supplementary Fig. 3a-c, suggesting that our assumption is not unrealistic.

It is our understanding that this concern of the reviewer might stem from our misuse of the term “bounded evidence accumulation” in the original manuscript. As we detail in the later concern about Ornstein-Uhlenbeck processes, although we assume a stimulus is consciously perceived when the accumulated evidence reaches the decision bound, we do not assume that evidence accumulation stops immediately after.

We now mention that the modelling approach requires post-decisional evidence accumulation, and discuss this issue, acknowledging other possible models of confidence:

Lines 354-364:

To achieve this confidence readout, we relied on two modelling assumptions: that evidence accumulation i) continues after reaching the bound and ii) leaks information. Indeed, LIP neurons documented in non-human primate studies show a sharp drop in firing rate immediately after the decision, consistent with an absorbing bound, implying that evidence accumulation stops after a decision. Such a mechanism would lead to constant maximal evidence for “yes” responses, and therefore require a second accumulator towards the opposite choice to derive graded confidence ratings (Kiani & Shadlen, 2009, Science). This second accumulator may be represented by the neurons we found showing a higher increase in firing rate for misses compared to hits, raising also the intriguing possibility that misses are encoded actively. This extends a study on non-

human primates which found prefrontal neurons also coding for misses but rather in the delay period following the stimulus (Merten & Nieder, 2012, PNAS). However, more work is needed to show how such neurons could contribute to confidence in detection tasks.

L47 "...evidence accumulation process reaches a threshold [12] similar to the physiological processes underlying decision-making.[13-16]" Reference 40 is germane to both parts of this sentence. Reference 14 is irrelevant; it presents no support for evidence accumulation, nor is it a reaction time task. Causal evidence for the influence of parietal neurons on choice and RT can be found in Hanks et al, 2006 Nat Neurosci 9: 682 - 89.

We removed reference 14, following the reviewer's request. As for reference 40, we added the sentence "A behavioral modelling study proposed a similar mechanism for the subjective experience of reaching a decision (Kang et al., 2017, Cur. Bio.)" (Lines 41-42).

Fig 1B. The horizontal dashed line labeled "no" is not described in the figure caption. How does the subject indicate "no"? I was led to believe it was just the absence of a key press by 2 s. The caption to B is confusing overall. More details are needed in Methods about the behavior. For example, what is the operational definition of a false alarm? Would a response at 2.1 s count as a FA or a Miss? Or are there sham trials that are interleaved? If so, does the subject just experience these as longer gaps? Was there feedback given? If so was the or correct and errors? If so, when was the feedback given for a miss and correct reject?

We have modified Fig. 1b, as well as its legend. We now dedicate Fig. 1c to show the response times for hits with the 2 s deadline. As the reviewer can judge, all response times were well under 2 s so we do not think that this is a problem. We do however agree that one sentence in the results was unclear and we changed it to:

Lines 90-91: The participant rarely responded outside this 2 s window (0.36%; false-alarms)

No sham trials were interleaved. Inter stimulus interval was defined as a constant 2 s delay plus an exponential distribution with a mean of 2 s, therefore the participant experienced gaps of different lengths. No feedback was given to the participant.

Fig. 1 (b) The participant pressed a key as soon as they felt a stimulus. In this example, the first stimulus is a miss (i.e. no key press within 2 s following stimulus onset) and the second stimulus is a hit (i.e. key press within 2 s following stimulus onset). ISI: inter-stimulus interval. **(c)** Distribution of response times (RT) and response deadline for a trial to be considered as a hit (red vertical dashed line)

L62 “Yet, to our knowledge the underlying neural mechanisms remain to be described.” See Kiani & Shadlen, Science 2009.

This sentence was unclear, as we meant to refer to the neural mechanisms of confidence in detection tasks, which to our knowledge is indeed unknown. We modified the sentence to “Yet, to our knowledge, the underlying neural mechanisms of confidence in detection tasks remain to be described”.

L103 The statement about covariance is weak. Many processes exhibit a decrease in correlation as a function of time lag. For diffusion one must also observe an increase in correlation, for fixed lag, as a function of time. See major concern B.

Fig 1E shows the increase in VarCE up to 1 second. That can be explained by a stopping bound.

These two points should not be a concern in the revised version of the manuscript which does not include variance analyses.

L187. Experiment 3 is a nice addition. It shows that neurons (some parietal neurons) process the stimulus even when the subject is not engaged in the task. But i don’t understand what the term “task activity” means. Also the lack of a report does not mean the brain did not prepare one provisionally.

We agree that the use of “task activity” was inadequate in the context of a no-report paradigm. Instead, we now refer to “*activity related to reporting rather than perceptual processing leading to consciousness*”. (Lines 201-202) We also added an introductory sentence to Experiment 3: “*While the use of delayed responses in Experiment 2 ensured that the comparison of hits and misses was not contaminated by motor confounds, a possibility remains that the neurons we uncovered are involved in task execution rather than subjective experience per se.*” (Lines 190-192)

Regarding the reviewer’s second point, it is true that we have no quantitative way of assessing whether the patient was covertly performing the task. As noted by reviewer 3, this concern is also related to a critique of no-report tasks formulated by N. Block (2019, TiCS).

L212 “A stimulus was perceived if the simulated evidence accumulation (EA) process reached a bound 40 at any time during the stimulus window (Fig. 4A), compatible with all-or-none views of conscious access.” This could be said of non-conscious processing too. For example, something flickers in the visual periphery. The parietal cortex may or may not encode a plan to direct attention there. The subject may be unaware but subsequent eye movements might reveal that the object was effectively interrogated. Such gnostic states probably rely on the provisional commitment that makes use of a threshold applied to evidence, but it need not result in consciousness.

The reviewer seems to interpret our claims regarding EA as necessary and sufficient for consciousness. As we clarified above, by no means do we believe that all EA processes reaching a bound result in a conscious percept. Indeed, the example provided by the reviewer may well involve EA reaching a bound too (although this is a fairly complex situation, that may be qualified as preconscious according to an influential taxonomy of conscious states, see Dehaene et al., 2006). Yet, our model predicts decisions of reporting a consciously perceived stimulus, and we were careful to avoid typical confounds using delayed-report and no report paradigms. We added a sentence to clarify evidence accumulation may provide a necessary, if not sufficient mechanism for consciousness.

“Furthermore, conceptual work will be needed to assess to what extent evidence accumulation is necessary if not sufficient for consciousness [...]”. (Lines 410-413)

L226 Fig 4D, Why is confidence high on the Misses?

Please see our response above.

The use of the Ornstein-Uhlenbeck process (OUP) in Experiment 4 is confusing. The OUP is not really an evidence accumulation process; it is a leaky accumulation, more like smoothing. Was

this used only for the ECoG or do the authors think the electrophysiology in Experiment 1 is also described by such a model? If so the VarCE and CorCE relationships associated with accumulation are misguided. Presumably the author believe the same mathematical process underlies the decisions in both experiments. If so, then the prediction for confidence should not be based on a signal that does not represent the underlying process.

We first take this opportunity to clarify that by “Ornstein-Uhlenbeck process”, we meant a continuous leaky evidence accumulation process that may or may not reach a bound (that we relate to conscious perception). We reasoned that this process could continue after the decision, but not forever, thus the decay in drift rate was added to the model. The leakage was a practical way of driving the decision variable (DV) to zero after the decision and allowed us to fit the shape of the average DV to the EEG data. With this in mind, we argue that it is still an “evidence accumulation process”, even if it is not perfect accumulation. While reading our manuscript, we realized that we wrongly used the term “bounded evidence accumulation” twice and apologize for this mistake which we have corrected. We now completely dropped the terms “Ornstein-Uhlenbeck” and “bounded evidence accumulation” and only write about “evidence accumulation” or “leaky evidence accumulation”. We are of course willing to change this if the reviewer proposes a better taxonomy.

We agree with the reviewer that the leakiness and post-decisional evidence accumulation assumptions of our model need to be clarified with respect to the neuronal data. As we briefly introduced in an earlier response, our take is that both the neuronal data and the computational model of confidence are consistent with leakiness and post-decisional evidence accumulation. In what follows, we present some evidence that our modelling assumption can simulate evidence accumulation traces that are qualitatively similar to the firing rates we observed. However, we would rather refrain from making strong claims about the parameters of the underlying neuronal dynamics in the manuscript, as this issue is –in our view– too complex to address in the present manuscript.

In experiment 1, the firing rates of the neurons after the button press did not drop immediately, as in LIP studies (e.g. Roitman & Shadlen, 2002, J. Neuro.). Instead, they briefly continued to increase, reaching a higher maximum for shorter RTs. We now present these data in the revised manuscript (see Fig 1f (inset); Supplementary Fig. 3). This is consistent with our model. To illustrate this, we simulated the DV using our model and separated the simulated DVs into misses and three bins of RT for hits. We were able to find parameters that reproduced patterns similar to the neuronal data.

Supplementary Fig 8. Comparison between neuronal data and evidence accumulation traces simulated by the computational model. Left: Average firing rates for detection- and RT- responsive neurons, for three bins of RT (hits; blue) and for misses (red). Firing rates were normalized to a 0.3 s pre-stimulus baseline and time-locked to the stimulus onset. Inset: Corresponding average firing rates time-locked to the response (data identical to Fig. 1F, displayed here for comparison purposes). Right: Average evidence accumulation traces, time-locked to the stimulus onset, as simulated by the model for a set of manually selected parameters. Inset: Corresponding average evidence accumulation traces time-locked to the response time of the model. To make the simulation more realistic, we separated the non-decision time into a perceptual component (corresponding to the delay for perceptual evidence to start being accumulated) and a motor component, corresponding to the delay between the decision and the motor response. We set the motor delay to 80 ms (Resulaj et al., 2009, Nat. Neuro) and added a small amount of variability to both components (Gaussian random noise with a standard deviation of 40 ms).

The inclusion of a leakage parameter departs from close-to-perfect accumulation during discrimination tasks using random-dot kinematograms documented both at the behavioral (Kiani et al., 2008, J. Neuro) and neural levels in the LIP (Churchland et al., 2011). However, we hope the reviewer agrees that for first-order responses in detection tasks using short vibrotactile stimuli with unpredictable onsets, the brain may adapt the dynamics of evidence accumulation to fit the task. Indeed, without leakage, neurons would need to “know” when to start accumulating evidence, otherwise, arguably, the state of the accumulator would increase with time prior to the stimulus onset, as noise is integrated. Other works using change detection tasks support our speculation (e.g. Ossmy et al., 2013, Cur. Bio.), although we are also aware of some works that

do not (Waskom & Kiani, 2018, *Cur. Bio.*). Evidence accumulation prior to stimulus onset is supported by evidence from a scalp EEG study (Devine et al., 2019, *eLife*).

For confidence, post-decisional readouts are well accepted (as argued in the response to the corresponding concern) and one modelling study additionally found evidence supporting leakage (Yu et al., 2015, *J. Exp. Psych.*). We are not aware of any evidence accumulation model that describes confidence in a detection task, especially with unpredictable stimulus onsets.

In sum, our modelling choice of using leakage:

- i) Provided a way to model the decrease of the EEG signal (and firing rates).
- ii) Is more adapted to a detection task with short stimuli and unpredictable stimulus onset, as shown by some prior works (Ossmy et al., 2013, *Cur. Bio.*)
- iii) Provides a parsimonious way of computing confidence in such a setting.
- iv) Allows us to reproduce similar evidence accumulation traces as the firing rates when time-locking to the response onset.

As for any model, ours does not provide a definitive answer concerning the general mechanism underlying confidence ratings, but we hope that our arguments and toy examples convinced the reviewer that leakiness is not an unreasonable assumption and that a definitive answer on this much debated matter is out of the scope of the present manuscript. That said, we thank the reviewer for raising this crucial point that we completely overlooked in the original manuscript and that was causing confusion. We have now added two paragraphs on the topic in the discussion (lines 354-380) and included the Figure above as Supplementary Figure 8.

With respect to the variance analysis, we had a modest aim: showing an increase in VarCE and a decrease in CorCE with increasing lag to support our evidence accumulation interpretation. We did not presume that our data would allow us to dissociate between leaky and ideal accumulators. As we have dropped this variance analysis (see answer to concern B), this should not be a concern anymore.

Metacognitive sensitivity (LL 798-802). Please spell out what you calculated and how the numbers relate to the values reported in the text and graphs. It is reasonable to express skepticism about the metacognitive measures (e.g., meta-dprime) because they fail to acknowledge that the distribution of S+N is not the same as the conditional distribution of S+N, given the response. This creates distortions which are further compounded by violations of the Gaussian and variance assumption.

We used the area under the receiver operating characteristic (AUROC) curve (Fleming et al., 2011, *Science*). This measure thus “decodes” first-order performance using confidence ratings. If participants always rate their confidence at 100% when they are correct and at 50% when incorrect, the AUROC will be 1. In the other extreme case, if participants give random confidence ratings, the AUROC will be 0.5. We would like to point to the reviewer’s attention that the AUROC measure does not rely on any statistical assumptions and does not suffer from the limitations of meta-dprime. Its main criticism has been that it does not compensate for differences in first-order performance. In our manuscript however, we never attempt a comparison between AUROC measures for “yes” and “no” responses, nor between participants. We use AUROC measures to show that our model reproduces the metacognitive abilities of the participants. As our model reproduced first-order performances well (hit-rate and false-alarm rate; see Supplementary Fig. 6), our analysis is not affected by this issue.

We clarified the corresponding section of the methods:

We evaluated metacognitive sensitivity or how well confidence predicted task performance (Fleming & Lau, 2014). For this, we represented the posterior probability of confidence ratings knowing the correctness of the detection report using the area under the receiver operating characteristic (AUROC) curve, computed independently for “yes” responses (hits and false alarms) and “no” responses (correct rejections and misses). (Lines 827-820)

The authors seem to be unaware of older work from David Heeger using fMRI to study a similar detection task (e.g., Ress et al., *Nat Neurosci* 2000).

We included this reference in the introduction (lines 42-45): “*Various neuroimaging studies have interpreted increases in neural activity elicited by detected stimuli (Ress et al., 2000, Nat Neuro.) (versus missed stimuli) as evidence accumulation (Salti et al., 2015, eLife; Tagliabue et al., 2015, Sci. Rep.; Wyart & Tallon-Baudry, 2009, J. Neuro)*”

Reviewer #3 (Remarks to the Author):

Pereira et al. present an elegant set of complementary experiments, including intracranial single neuron and ECoG recordings from a single patient, and scalp EEG in a group of 18 healthy subjects. They suggest that their experimental and computational model evidence support a neural evidence accumulation account of “perceptual consciousness” and “perceptual monitoring” of a vibrotactile stimulus presented at near-detection level.

Overall, I feel that this paper is technically and conceptually very impressive. The experimental design and data analysis approaches are highly rigorous and take care to appropriately control for important factors (e.g. using delayed and no report paradigm) to support the major findings. The paper is an important contribution to our understanding of mechanisms that give rise to conscious access, importantly providing a computational framework that could be further extended in the future to test generalizability (beyond the multiple experiments already presented here).

We thank the reviewer for their appreciation of our work.

My concerns are only minor. I found the narrative to be sometimes difficult to follow, and I think improvements are needed in the flow and the consistency of terminology used. Moreover, the authors largely describe the single neuron, ECoG, and EEG analyses as replications of one another, but these distinct modalities do not capture the same neural phenomena. I hope that the authors can address the following points:

- I think the manuscript title should be changed to “Evidence accumulation determines VIBROTACTILE conscious access” to better acknowledge the scope of the paper

We agree with the reviewer that our title was not specific enough, although looking at our references, we also note that very few authors specify the sensory modality in the title. Instead of putting the focus on the vibrotactile nature of our stimuli, we would like to emphasize that evidence accumulation concerns both perceptual consciousness and perceptual monitoring. Therefore, if both the editor and reviewer agree, and in line with the simplified writing we now use throughout the text, we propose to change the title to “*Evidence accumulation determines perceptual consciousness and monitoring*”. We now suggest in the discussion that evidence accumulation may generalize to other perceptual domains beyond vibrotactile stimuli, although further research will be necessary to prove it.

- The ECoG and EEG results are largely described as “replication” or “generalization” of the single neuron findings. Yet in both ECoG and EEG analyses, the authors focus on raw voltage fluctuations, which can reflect electrophysiological processes that are unrelated to neuronal firing. In the domain of ECoG, evidence suggests that high-gamma/high frequency broadband (~50-200 Hz) power amplitude may be more closely related to neuronal population firing near a given electrode contact (see Parvizi & Kastner 2018 Nat Neurosci for recent review). The authors could consider redoing the ECoG analyses based on high-frequency power amplitude. At minimum, the issue should be discussed, and the analyses should not be framed as “replication” (though the similarity of findings across neuron firing, ECoG and EEG is indeed striking as presented).

It is true that although we obtained comparable results across recording methods, the underlying physiological processes may differ. We now refrain from using the term “replication”, and acknowledge in the discussion that ECoG/EEG and single-unit results were based on arguably distinct electrophysiological processes:

Line 331-333: Analyzing the amplitude of ECoG data and EEG data from Experiment 4, we could generalize our single-neuron findings at a larger scale and arguably distinct electrophysiological processes.

We performed the same analysis for Experiment 1 using broadband gamma (50 – 180 Hz). To compensate for the 1/f drop, we filtered the data in consecutive 10 Hz sub-bands. For each sub-band, we stored the absolute value of the Hilbert transformed filtered data, divided by its mean value. We then averaged sub-bands to get the broadband gamma signal. The reviewer can see the result for broadband gamma below (a), for the electrode depicted in Figure 1i (revised manuscript), which we show in (b) for the reviewer’s convenience. Although the broadband data shows a buildup prior to the RT, unlike the low frequency ECoG data, the final amplitude seems to decay with time.

We would like to point out however, that a recent paper recording laminar data in monkeys suggests that ECoG electrodes measure the broadband activity of superficial sources for which no clear corresponding multi-unit activity can be found, suggesting that these broadband gamma signals could reflect dendritic processes rather than neuronal activity (Leszczyński et al. 2020, *Sci. Adv.*).

Therefore, considering i) the uncertain link between broadband gamma and neuronal activity, ii) the quantity of results already contained in this manuscript and iii) the fact that the data is open anyways, we prefer to not include these results in SI unless the editor / reviewer argues otherwise.

(a) Average broadband gamma, aligned to stimulus onset from one electrode posterior to the microelectrode array for three terciles of RT (blue) and for misses (red). (b) Average ECoG amplitude, for the same electrode and time-locking. In all panels, shaded areas represent 95%-CI.

- In the abstract, the first sentence describes “perceptual consciousness” as encompassing (1) the perceptual experience, and (2) reflexive monitoring. Later, these terms are described as “perceptual and reflexive consciousness” and then throughout the paper, the term “perceptual monitoring” is used (I believe synonymously with reflexive monitoring). The title of the paper refers to “conscious access.” The paper would be much easier to follow if terms are used much more consistently throughout. Relatedly, I recommend referring to the following paper, which distinguishes between access and phenomenal consciousness Block, N. (1995). On a confusion about a function of consciousness. Behavioral and brain sciences, 18(2), 227-247.

We agree that our terminology was not consistent enough. We rewrote parts of the abstract and introduction based on the reviewer’s comment, and simplified our terminology accordingly. We now use only “perceptual consciousness and monitoring” in the title, abstract and introduction, which refer respectively to subjective experience and metacognitive monitoring. We also removed all instances of the term “conscious access”, to avoid any ambiguities regarding phenomenal vs. access consciousness as put by Ned Block.

- Abstract: “...irrespective of response effectors” is ambiguous here without context (though I understood after reading the whole paper) and I think should be worded more clearly

We agree that this sentence is unclear. We changed it to “irrespective of motor confounds”.

- The rationales for the three experimental paradigms (immediate response, delayed response, and no report) is not really fully explained until late in the Discussion (second last paragraph, lines 369-372). The use of these multiple paradigms is a major strength of this study, and I think the rationales should be further clarified and explicitly stated in the Intro and better embedded throughout the description of results. In the Intro, a short discussion of how correlative (reported vs. unreported) findings have been suggested to be confounded by postperceptual processes would be helpful to motivate their exp. 2 and exp. 3. I recommend referring to the following paper: Block, N. (2020 Trends Cogn sci). Finessing the bored monkey problem.

The reviewer is right in that we did not motivate our different paradigms early enough in the manuscript. We now mention this in the introduction:

Lines 72-75:

Namely, the comparison of neural signals between paradigms involving immediate and delayed responses ensured that the NCCs we isolated were not contaminated by motor actions. Moreover, the use of a no-report paradigm ensured that these NCCs reflected perceptual consciousness per se, irrespective of task demand.

We now also provide a clearer narration in the results section when introducing Experiment 2:

Lines 133-134: Because the participant hit a key as soon as they detected a stimulus in Experiment 1, the comparison between hits and misses mentioned above may be contaminated by motor confounds. Thus, we tested whether neuronal responses relate to conscious perception irrespective of motor actions by imposing a delay between stimulus onset and reports.

Introducing Experiment 3, we now recapitulate the advantage and limitations inherent to Experiment 2, before resuming to the original description of the no-report paradigm we wrote previously:

Line 190-192: While the use of delayed responses in Experiment 2 ensured that the comparison of hits and misses was not contaminated by motor confounds, a possibility remains that the neurons we uncovered are involved in task execution rather than subjective experience per se. Finally, when summarizing our results at the beginning of the discussion, we now remind the reader that “Using a combination of delayed response and no-report paradigms, we ruled out the possibility that these effects stemmed from motor actions or task demand” (lines 290-292).

We thank the reviewer for helping us improving our rationale throughout the revised manuscript.

- An updated review on global neuronal workspace perhaps could be cited when discussing relevant concepts: Mashour, G. A., Roelfsema, P., Changeux, J.-P., & Dehaene, S. (2020). Conscious processing and the global neuronal workspace hypothesis. *Neuron*, 105(5), 776-798.

We thank the reviewer for pointing this out. Although we were aware of this updated version, it somehow slipped from the submitted manuscript. This is now corrected.

- In the introduction, I recommend that the authors expand on the notion of ignition and the role of evidence accumulation in GWT, as the entire paper is built from these theoretical considerations. To my knowledge, the concept only applies to GWT and the authors should be explicit about this (or argue otherwise). Higher order models of consciousness and recurrent activation theories do not suggest such a mechanism (e.g. Lau, H. (2019b). *Consciousness, Metacognition, & Perceptual Reality Monitoring*. PsyArXiv. doi:10.31234/osf.io/ckbyf).

We agree with the reviewer that ignition and evidence accumulation are key mechanisms put forward by GWT. At the same time, establishing a clear formal link with GWT and ruling out a possible role of evidence accumulation in other theoretical frameworks is beyond the scope of the present study. At the current stage, and as we indicated in our response to reviewer 2, we consider evidence accumulation to be agnostic with respect to which specific theoretical framework may explain consciousness. To make this clearer, we now state that *“Furthermore, conceptual work will be needed to assess to what extent evidence accumulation is necessary if not sufficient for consciousness (Vlassova, 2014, PNAS; Barbosa et al., 2017, Consc. & Cog.), and how it fits within other theoretical frameworks such as higher order thought theories (Brown et al., 2019, TiCS) or recurrent processing theories of consciousness (Lamme, 2005, TiCS).”* (lines 410-413). Unless the reviewer disagrees, we will cite the TiCS opinion paper from the same authors, which provides a more general description of higher order theories.

- For the EEG study, it seems that details may be missing regarding how subjects were screened inclusion/exclusion criteria.

We added the following sentence in the methods section (lines 622-623):

All participants reported being right-handed, had normal hearing and normal or corrected-to-normal vision, and no psychiatric or neurological history. Two participants were excluded due to excessive artifacts.

- If possible, the authors should report on the intracranial patient's state at the times of testing (e.g. medications, pain, cognitive state/mood), especially since results are based on a single case.

This is important and has been added to the original manuscript in the methods section (lines 615-619):

Except for ibuprofen, paracetamol and esomeprazole, the patient was not under the influence of any medication (i.e., no antiepileptic drug). The patient rated their pain at 1 out of 10 on a visual analog scale, except for Experiment 3, for which they rated their pain at 2 out of 10, reporting a slight pain around the left orbit. No neurological or cognitive abnormality, or anxiety were reported by the clinical team and their mood was stable.

- Please add information about the size of ECoG electrode contacts

We added the following sentence to the methods section (line 678):

The electrodes had a 4 mm diameter with 2.3 mm exposed corresponding to an area of 4.15 mm²

- In Fig 1D, the "Slow/Mid/Fast" text is partly cut off

This has been corrected.

REVIEWER COMMENTS

Reviewer #1 (Remarks to the Author):

The authors have provided a comprehensive response to my comments. I was struck by how delayed/extended the response of these newly characterised “miss”-coding neurons was, even in Experiment 1. It’s up to the authors and editor of course, but if similar extended responses in the delay period of Experiment 2 are seen this would seem informative to include in the supplement.

One minor final suggestion - in the new lines 350-352, perhaps “irrespective of detection” should be “irrespective of detection response”, given that what matters here is the sign of the response (yes or no), not the task itself.

Reviewer #2 (Remarks to the Author):

The authors have performed a very thorough revision. I commend them for the serious thought they put into this revision. I was enthusiastic about the original submission and all the more so now. I have no further recommendations.

Reviewer #3 (Remarks to the Author):

I am satisfied with the authors' response and congratulate them on an outstanding manuscript. I am ok with the authors leaving out the ECoG broadband gamma result given that they now acknowledge the distinct processes captured by single-unit and ECoG/EEG raw voltage. Though the Leszczynski 2020 paper noted by the authors concludes that broadband gamma does not always correspond to multi-unit activity, I encourage them to further examine the findings from that paper, which does actually report a strong temporal correlation between spiking and broadband gamma (in line with lots of other literature). The dissociation that they report is very context-specific (and potentially region specific). This is just a note for future reference, and again, I congratulate the authors on their paper which I think is acceptable for publication in its current form.

Reviewer #1 (Remarks to the Author):

The authors have provided a comprehensive response to my comments. I was struck by how delayed/extended the response of these newly characterised “miss”-coding neurons was, even in Experiment 1. It’s up to the authors and editor of course, but if similar extended responses in the delay period of Experiment 2 are seen this would seem informative to include in the supplement.

One minor final suggestion - in the new lines 350-352, perhaps “irrespective of detection” should be “irrespective of detection response”, given that what matters here is the sign of the response (yes or no), not the task itself.

Response: Like the reviewer, we were intrigued by the shape and latency of the responses we obtained from miss-coding neurons. As we now mention in the revised manuscript, we found no miss-coding neurons in Experiment 2, possibly because there were five times less detection-selective neurons (17 instead of 81).

We changed “irrespective of detection” by “irrespective of detection response”.

We thank the reviewer for her/his kind words and invaluable feedback.

Reviewer #2 (Remarks to the Author):

The authors have performed a very thorough revision. I commend them for the serious thought they put into this revision. I was enthusiastic about the original submission and all the more so now. I have no further recommendations.

Response: We thank the reviewer for her/his kind words and invaluable feedback.

Reviewer #3 (Remarks to the Author):

I am satisfied with the authors' response and congratulate them on an outstanding manuscript. I am ok with the authors leaving out the ECoG broadband gamma result given that they now acknowledge the distinct processes captured by single-unit and ECoG/EEG raw voltage. Though the Leszczynski 2020 paper noted by the authors concludes that broadband gamma does not always correspond to multi-unit activity, I encourage them to further examine the findings from that paper, which does actually report a strong temporal correlation between spiking and broadband gamma (in line with lots of other literature). The dissociation that they report is very context-specific (and potentially region specific). This is just a note for future reference, and again, I congratulate the authors on their paper which I think is acceptable for publication in its current form.

Response: We agree with the reviewer that the relationship between gamma and multi-unit activity is complex and this specific study by Leszczynski and colleagues does not settle the issue. We will keep this in mind for the future.

We thank the reviewer for her/his kind words and invaluable feedback.